# Nup107 is a crucial regulator of torso-mediated metamorphic transition in *Drosophila melanogaster*

**Jyotsna Kawadkar\*, Pradyumna Ajit Joshi, Ram Kumar Mishra\***

Department of Biological Sciences, Indian Institute of Science Education and Research (IISER) Bhopal, Bhopal, India

## eLife Assessment

This **valuable** study presents findings on the developmental roles of Nup107, a key nucleoporin, in regulating the larval-to-pupal transition in *Drosophila melanogaster* through its involvement in ecdysone signaling. The evidence supporting the authors' claims is **solid**, with robust experimental approaches including RNAi knockdown and rescue experiments. The authors propose that Nup107 influences EcR localization indirectly by reducing the expression of Halloween genes, a consequence of impaired Torso signaling. However, it remains uncertain whether Torso is the sole receptor tyrosine kinase involved, and this disruption ultimately leads to decreased ecdysone production. In addition, finding a mechanism would strengthen the findings as the currently proposed mechanism is not completely supported by the data.

**Abstract** Nuclear pore complexes (NPCs), composed of nucleoporins (Nups), affect nucleocytoplasmic transport, thus influencing cell division and gene regulation. Nup107 subcomplex members have been studied in housekeeping functions, diseases, and developmental disorders. We report a unique regulatory function for Nup107 in metamorphic transition during *Drosophila* development. RNA interference (RNAi)-mediated *Nup107*-depleted larvae were arrested in the third-instar larval stage with no signs of pupariation. This lack of pupariation is primarily due to inhibited nuclear translocation and transcriptional activation by EcR. We demonstrate the involvement of Nup107 in the transcription of the *Halloween* genes, modulating ecdysone biosynthesis and the EcR pathway activation. The regulation of EcR-mediated metamorphosis by the receptor tyrosine kinase, *torso*, is well documented. Accordingly, overexpression of the *torso* and MAP-kinase pathway activator, *ras^{V12}*, in the *Nup107* depletion background rescues the phenotypes, implying that Nup107 is an epistatic regulator of Torso-mediated activation of EcR signaling during metamorphosis.

## Introduction

The multi-protein assembly of nucleoporins (Nups) constitutes nuclear pore complexes (NPCs). In eukaryotic cells, NPCs serve as molecular conduits for the trafficking of proteins and RNAs between the nucleus and the cytoplasm. The NPC composition, as ascertained from proteomic analyses conducted in yeasts and vertebrates, has contributed to our understanding, revealing that approximately 30 distinct Nups are arranged in sub-structures and are distributed on different faces of NPCs (*Cronshaw et al., 2002*; *Rout et al., 2000*). The substructures are the outer cytoplasmic and inner nuclear rings, which surround a central inner ring called the scaffold ring, primarily aiding in the process of nucleocytoplasmic transport (*Lin and Hoelz, 2019*).

**\*For correspondence:**
jyotsna19@iiserb.ac.in (JK);
rkmishra@iiserb.ac.in (RKM)

**Competing interest:** The authors declare that no competing interests exist.

**eLife digest** Fruit flies are a widely used model organism for genetic and developmental studies because of their genetic similarity to humans. One example is the study of metamorphosis, the process during which flies develop from eggs into larvae, pupae, and ultimately adults. A comparable process in mammals is puberty, when juveniles mature into fully developed adults.

Puberty involves profound physical changes, such as the attainment of sexual maturity and the emergence of new social behaviors. These major life transitions are primarily regulated by hormonal signals from the brain, ovaries and adrenal glands. Imbalances in these hormones can delay or disrupt pubertal development. However, the underlying mechanisms remain incompletely understood.

To address this gap, Kawadkar et al. used established genetic tools to reduce or eliminate the nuclear pore complex protein Nup107 in fruit flies. This protein is essential for the movement of molecules between the nucleus and the surrounding space, the cytoplasm.

Reduced levels of Nup107 decreased ecdysone production, the steroid hormone needed to start metamorphosis. As a result, the development of fruit fly larvae was disrupted, with animals failing to progress efficiently to the pupal stage. Kawadkar et al. further showed that the ecdysone receptor did not properly move into the nucleus, where it would activate specific genes necessary for metamorphosis. This prevented gene expression essential for developmental progression.

The findings of Kawadkar et al. suggest that Nup107 may have a broader role in developmental processes that depend on steroid hormones. Indeed, in humans, Nup107 mutations are known to disrupt gonad development. In insects, the production of ecdysone is indirectly affected by Nup107. Supporting this, feeding synthetic ecdysone to Nup107-depleted larvae partially restored metamorphosis, allowing the animals to reach the pupal stage. Similarly, overexpressing the torso gene, which is part of the signalling pathway that stimulates ecdysone production, fully rescued timely metamorphosis, suggesting that activating the hormone pathway can compensate for the defects caused by reduced Nup107 levels.

Overall, these results clarify how Nup107 controls the production of the steroid hormone ecdysone at the start of metamorphosis in fruit flies. This work opens new avenues for investigating whether Nup107 performs similar roles during steroid hormone-dependent puberty in humans. Furthermore, these findings suggest that Nup107 may regulate broader neuroendocrine functions in the brain, with wide-ranging implications for the development of an organism.

The largest Nup107 sub-complex, also called the Y-complex, is symmetrically located on both sides of the nuclear membrane. In metazoans, the Y-complex is composed of 10 distinct Nups (ELYS, Nup160, Nup133, Nup107, Nup96, Nup85, Nup43, Nup37, Sec13, and Seh1). ELYS is a sub-stoichiometric Y-complex member and is present only on the nucleoplasmic side (*D'Angelo and Hetzer, 2008*; *Morchoisne-Bolhy et al., 2015*). Nup107, along with Nup133, forms the stalk of the Y-complex and is critically required for the Y-complex stability. The Nup107 complex plays a pivotal role in facilitating ELYS-coordinated post-mitotic NPC assembly (*Boehmer et al., 2003*; *Walther et al., 2003*) and messenger RNA (mRNA) export (*Baï et al., 2004*). Additionally, the Nup107 complex members actively participate in mitosis, contributing to the regulation of kinetochore-microtubule polymerization (*Mishra et al., 2010*; *Zuccolo et al., 2007*). The multitude of cellular processes in which the Y-complex performs vital functions suggests it to be a central component in maintaining cellular homeostasis.

As a stable constituent of NPC, many Nups, including the members of the Y-complex, associate with chromatin and exert transcriptional regulation. Notably, interactions between active genes and Nups occurring predominantly within the nucleoplasm have been reported for dynamic Nups such as Nup98, ELYS, and Sec13 (*Capelson et al., 2010b*; *Kalverda et al., 2010*; *Kuhn et al., 2019*). In *Drosophila*, the dual Nup, ELYS, governs the development, and ELYS RNA interference (RNAi)-induced developmental defects are due to the reactivation of the dorsal (NF-κB) pathway even during the late larval stages (*Mehta et al., 2020*).

Nup107 is associated with actively transcribing genes at the nuclear periphery (*Gozalo et al., 2020*). The disruption of Nup107 in zebrafish embryos leads to significant developmental anomalies, including the absence of the pharyngeal skeleton (*Zheng et al., 2012*). Moreover, the biallelic

Nup107 mutations (D157Y and D831A) correlate well with clinical conditions such as microcephaly and steroid-resistant nephrotic syndrome (*Miyake et al., 2015*). In *Drosophila*, Nup107 co-localizes with Lamin during meiotic division, and Nup107 depletion perturbs Lamin localization, leading to a higher frequency of cytokinesis failure during male meiosis (*Hayashi et al., 2016*). Nup107 influences the regulation of cell fate in aged and transformed cells by modulating EGFR signaling and the nuclear trafficking of extracellular signal-regulated kinase (ERK) protein (*Kim et al., 2010*).

*Drosophila* undergoes elaborate metamorphosis initiated by the neuropeptide prothoracicotropic hormone (PTTH) (*Rewitz et al., 2013*). Bilateral neurons projecting into the prothoracic gland (PG), when stimulated by the PTTH, induce ecdysone production, which is subsequently released into the circulatory system for conversion by peripheral tissues into its active form, 20-hydroxyecdysone (20E) (*Johnson et al., 2013*; *McBrayer et al., 2007*; *Shimell et al., 2018*). The 20E binds to the ecdysone receptor (EcR), and the whole complex translocates into the nucleus and binds to chromatin to activate ecdysone-inducible genes (*Johnston et al., 2011*; *Kozlova and Thummel, 2002*; *Tennessen and Thummel, 2011*). In the PG, the primary neuroendocrine organ, PTTH signals through the receptor tyrosine kinase (RTK), Torso. The Torso-dependent activation of the MAP kinase pathway is responsible for the production and release of ecdysone hormone. However, ecdysone synthesis can also be regulated by the EGFR pathway (*Cruz et al., 2020*; *Yamanaka et al., 2013*). While Nup107 modulates EGFR pathway activation, the involvement of EGFR and torso pathways in ecdysone-dependent metamorphosis is undeniable. We dissect the involvement of Nup107 in Torso-mediated signaling and underlying mechanisms during *Drosophila* metamorphosis.

In a reverse genetic RNAi screening for Nup107 complex members, we noted that *Nup107* RNAi induces a significant developmental arrest at the third instar larval stage. Further analysis revealed that the EcR signaling pathway is perturbed, and EcR fails to translocate into the nucleus in *Nup107* knockdown. The failure of the EcR nuclear localization upon *Nup107* depletion is due to significantly reduced ecdysone hormone levels during the late third instar larval stage. Interestingly, overexpression of the *torso* and the *ras*^V12 in *Nup107*-depleted larvae rescued the developmental arrest and subsequently initiated pupariation. We propose that Nup107, an epistatic regulator of torso pathway activation in the PG, enables ecdysone surge for efficient metamorphic transition.

## Results

### *Nup107* is essential for larval-to-pupal metamorphic transition

Cell biological analyses in the mammalian cell culture system have shed light on critical regulatory roles of Nup107 in vertebrates. Yet its importance in development remains poorly understood. In this context, we started the characterization of *Drosophila* Y-complex member Nup107 in greater detail. Utilizing RNAi lines (*Nup107*^KK and *Nup107*^GD), we performed ubiquitous depletion through *Actin5C-GAL4*. Interestingly, *Nup107* depletion led to larvae arrest at the third instar stage, causing complete cessation of pupariation (120 hr after egg laying [AEL], right panel, *Figure 1A*), which was accompanied by an extension of larval feeding and growth periods. Quantitative assessment of *Nup107* transcript levels in the *Nup107*^GD and *Nup107*^KK RNAi lines suggested efficient *Nup107* knockdown (approximately 60–70%, *Figure 1B*). For further analyses, we generated polyclonal antibodies against the Nup107 amino-terminal antigenic fragment (amino acids 1–210; see Materials and methods for details). Purified anti-Nup107 polyclonal antibodies detected a band of approximately ~100 kDa in lysates prepared from the control larval brain complex. The intensity of this ~100 kDa band was significantly reduced in lysates prepared from organisms where *Nup107* was knocked down ubiquitously (*Figure 1C*) using *Actin5C-GAL4* driving *Nup107* RNAi (denoted as ubiquitous hereon). Further, the immunostaining with Nup107 antibodies identified a conserved and robust nuclear rim staining pattern overlapping with mAb414 antibodies recognizing FG-Nups and mRFP-tagged Nup107 expressed through its endogenous promoter (*Figure 1—figure supplement 1*) in salivary gland tissues. These observations confirm the efficacy of *Nup107* knockdown and provide a handle to assess levels and localization of Nup107 in affected tissues. To further investigate the role of Nup107 in development, we generated a *Nup107* null mutant using CRISPR-Cas9-mediated gene editing. This comprehensive approach involving RNAi-mediated knockdown and CRISPR-Cas9 gene editing is expected to provide valuable insights into the significance of *Nup107* in *Drosophila* development. The gRNAs targeting regions close to the start and stop codons of the *nup107* gene generated knockout

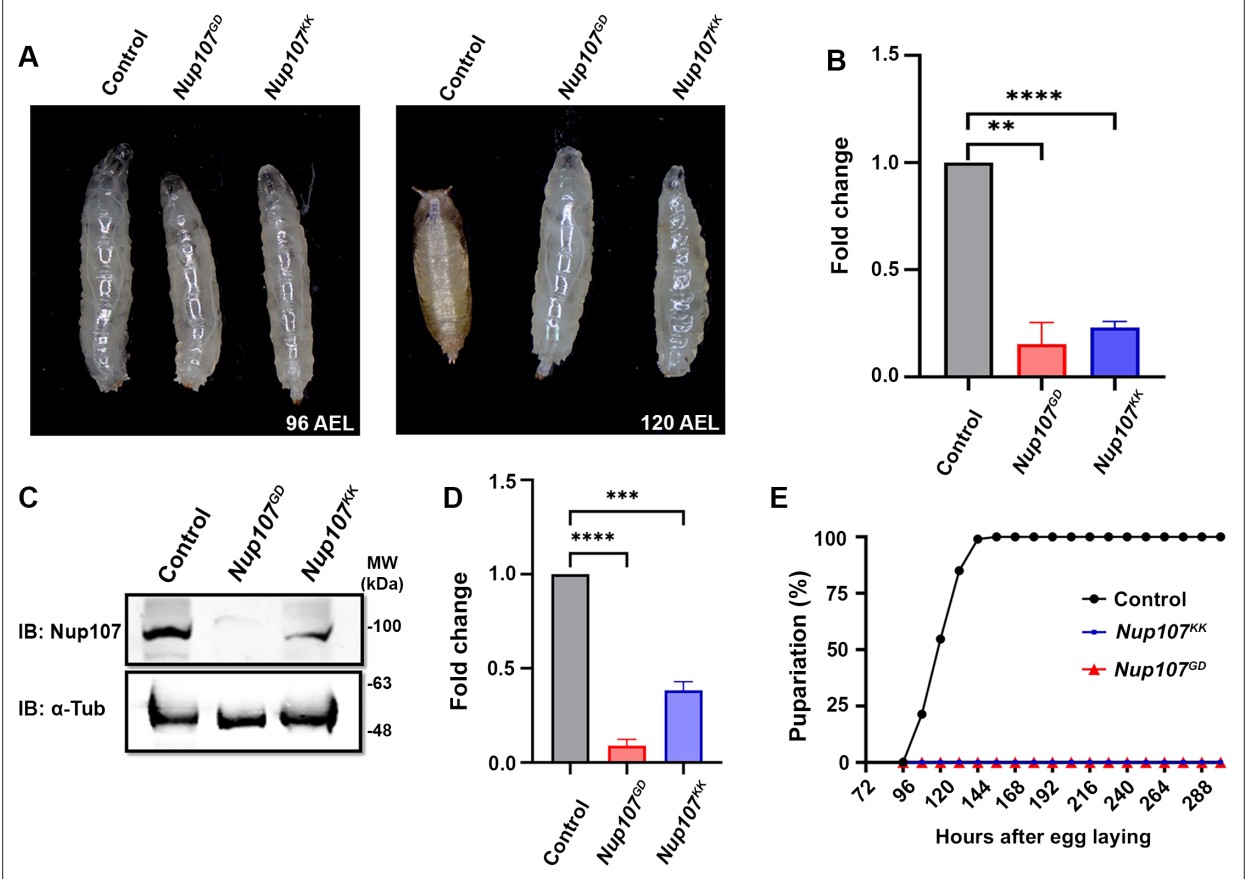

**Figure 1.** *Nup107* depletion impairs metamorphosis. Analysis of Nup107 depletion and its impact on growth and development of organism.
(**A**) Growth profile of third instar larvae from *Actin5C-Gal4*-driven control and *Nup107* knockdowns (*Nup107^{GD}* and *Nup107^{KK}* RNA interference [RNAi] lines) at 96 hr AEL (after egg laying) and 120 hr AEL. (**B**) Quantitation of *Nup107* knockdown efficiency. Data are represented from at least three independent experiments. Statistical significance was derived from the Student's t-test. Error bars represent SEM. **p=<0.001 and ****p=<0.0001. (**C**) Immunodetection of Nup107 protein levels in third instar larval brain-complex lysates from control and *Nup107* knockdown. (**D**) Quantification of Nup107 protein levels seen in (**C**). Data are represented from at least three independent experiments. Statistical significance was derived from the Student's t-test. Error bars represent SEM. ***p=<0.0002 and ****p=<0.0001. (**E**) Comparison of pupariation profiles of control and *Nup107* knockdown organisms.

The online version of this article includes the following source data and figure supplement(s) for figure 1:

**Source data 1.** Western blot analysis of Nup107 knockdown.

**Source data 2.** Original files for larval images and western blot analysis displayed in *Figure 1*.

**Source data 3.** Numerical values of graphs are shown in *Figure 1*.

**Figure supplement 1.** Nup107 staining in salivary glands.

**Figure supplement 1—source data 1.** Original confocal images are presented for *Figure 1—figure supplement 1*.

**Figure supplement 1—source data 2.** Original files for confocal images are displayed in *Figure 1—figure supplement 1*.

**Figure supplement 2.** *Nup107* CRISPR mutant generation.

**Figure supplement 2—source data 1.** The original DNA gel image corresponds to *Figure 1—figure supplement 2B*.

**Figure supplement 2—source data 2.** The raw original DNA gel image corresponds to *Figure 1—figure supplement 2B*.

($Nup107^{KO}$) mutants, which were confirmed by sequencing and PCR (*Figure 1—figure supplement 2*). However, the $Nup107^{KO}$ mutants could not be used as the $Nup107^{KO}$ homozygous shows lethality at the embryonic stage. So, we carried out all the analyses hereon with *Nup107* RNAi lines.

## *Nup107* contributes significantly to ecdysone signaling

The depletion of *Nup107* in *Drosophila* resulted in a distinct halt in growth and a developmental arrest at the third instar stage (*Figure 1A*). To discern the pathways affecting juvenile to adult developmental transition in *Drosophila*, we focused on levels of the sole insect steroid hormone, ecdysone.

We examined the localization of EcR in the late third instar salivary glands of both the control and *Nup107*-depleted larvae (using the $Nup107^{KK}$ line). Wild-type larvae exhibited normal EcR localization within the nucleus, but the nuclear translocation of EcR is perturbed in the *Nup107*-depleted larvae, with the bulk of the signal retained in the cytoplasm (*Figure 2A and B*). Quantitative analysis of EcR intensities in the cytoplasm and nucleus further established a significant decrease in EcR signals inside the nucleus upon *Nup107* depletion (*Figure 2C*). This observation suggests that Nup107 is required for EcR nuclear localization to mediate critical larval-to-pupal developmental transition. Furthermore, we noticed significantly smaller size salivary glands and the brain complex in *Nup107*-depleted larvae (*Figure 2—figure supplement 1*).

Prompted by the cytoplasmic accumulation of EcR, we investigated whether the mRNA levels of ecdysone-inducible genes were affected. Under normal conditions, 20E binds with EcR, and the activated EcR occupies the ecdysone response element in the promoter region of genes. The EcR is thought to function in a positive auto-regulatory loop, which may elevate EcR levels and sustain ecdysone signaling (*Varghese and Cohen, 2007*). First, we examined the EcR transcript levels, revealing a reduction under *Nup107* depletion conditions (*Figure 2D*). Subsequently, we measured the mRNA levels of known EcR target genes, *Eip75A* and *Eip74EF*, to find that the expression of each of these two target genes is reduced and correlates with *Nup107* knockdown levels (*Figure 2E and F*). Similar results were observed in the depletion of *Nup107* using the GD-based RNAi line (*Figure 2—figure supplement 2*). The defects observed during metamorphic transitions can be attributed to impaired ecdysone signaling in *Nup107*-depleted organisms, prompting a closer examination of their regulatory relationship.

## *Nup107* is dispensable for the nuclear import of EcR in the target tissue

We established that the nuclear localization of EcR is impaired upon *Nup107* knockdown (see *Figure 2B*). Typically, EcR nuclear localization follows an intricate mechanism where the 20E binding allows nuclear translocation of EcR and subsequent activation of its target genes (*Cronauer et al., 2007*; *Johnston et al., 2011*; *Lenaerts et al., 2019*). Biosynthesis of ecdysone hormone from dietary cholesterol is a prerequisite for EcR activation and requires a group of P450 enzymes coded by the *Halloween* genes (*Kannangara et al., 2021*; *Niwa and Niwa, 2014*). Moreover, the involvement of nucleoporin, Nup358, in facilitating nuclear transport of Met juvenile hormone receptors is well documented (*He et al., 2017*). Consequently, we posited the hypothesis that Nup107 either regulates active 20E-EcR complex formation by affecting 20E biosynthesis in ubiquitous *Nup107* knockdown scenarios or directly regulates EcR nuclear translocation in the target tissue.

We chose salivary glands to address these hypotheses and depleted *Nup107* using salivary gland-specific *AB1-GAL4* and PG-specific *Phm-GAL4*. Surprisingly, in contrast to the ubiquitous knockdown of *Nup107*, nuclear localization of EcR remained unaffected in salivary gland-specific *Nup107* knockdown (*Figure 3A and B* and *Figure 3—figure supplement 1*). Interestingly, the PG-specific *Nup107* knockdown phenocopied ubiquitous *Nup107* knockdown-induced EcR nuclear localization defects (*Figure 3C*, *Figure 3—figure supplement 1*). Quantification of EcR signals from *Nup107*-depleted late third instar salivary gland cells and comparison with control salivary glands indicates that the nuclear/cytoplasmic ratios are drastically reduced in case of PG-specific depletion but unaltered in salivary gland-specific *Nup107* depletion (*Figure 3D*, *Figure 3—figure supplement 1*). Consistent with these observations, expression levels of ecdysone-inducible genes *Eip75A* and *Eip74EF* were significantly reduced in PG-specific *Nup107* knockdown (*Figure 3E and F*, *Figure 3—figure supplement 1*). In accordance with unperturbed EcR nuclear translocation, salivary gland-specific depletion of *Nup107* yielded no discernible differences in larval growth and pupariation compared to the

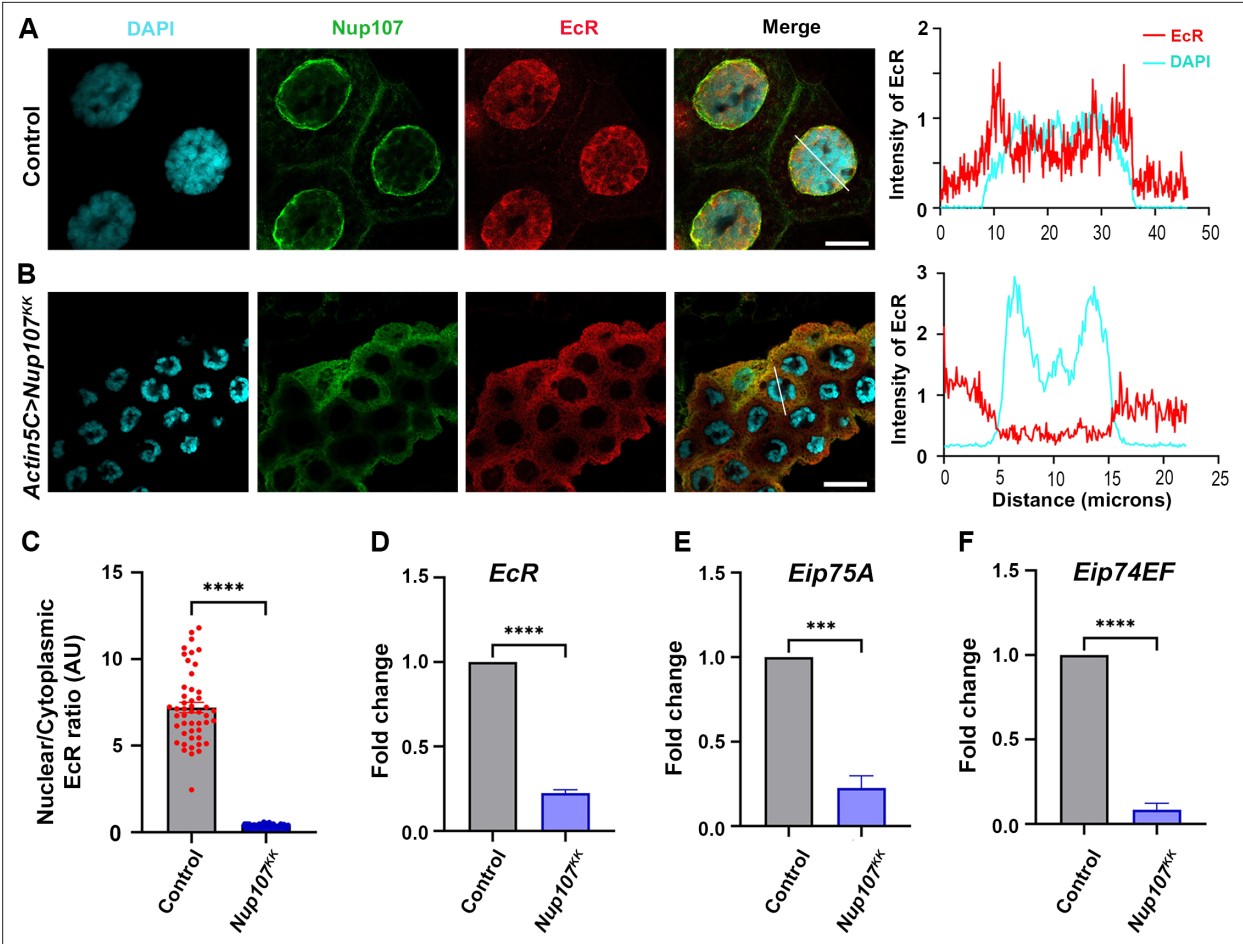

**Figure 2.** Ubiquitous knockdown of *Nup107* disrupts ecdysone signaling. Assessment of ecdysone receptor-dependent signaling in salivary glands of third instar larvae. (**A–B**) Staining of third instar larval salivary glands from control (**A**) and ubiquitous *Nup107* knockdown (**B**) with ecdysone receptor (EcR) (anti-EcR antibody, red) and Nup107 (anti-Nup107 antibody, green). DNA is stained with DAPI. Scale bars, 20 µm. Charts represent the line scan intensity profile of EcR (red) and DAPI (cyan) in the salivary gland nucleus region. (**C**) Quantification of the nucleocytoplasmic ratio of EcR under control and *Nup107* knockdown conditions. At least 45 nuclei were analyzed from seven to eight pairs of salivary glands. Statistical significance was derived from the Student's t-test. Error bars represent SEM. ****p=<0.0001. (**D–F**) Analysis of ecdysone-inducible genes, *EcR* (**D**), *Eip75A* (**E**), and *Eip74EF* (**F**) expression, respectively, at the onset of metamorphosis (late third instar larvae stage). Data are represented from at least three independent experiments. Statistical significance was derived from the Student's t-test. The error bars represent the SEM. ***p=<0.0004 and ****p=<0.0001.

The online version of this article includes the following source data and figure supplement(s) for figure 2:

**Source data 1.** Original confocal images for *Figure 2*, showing the EcR localization in Nup107-depleted tissues.

**Source data 2.** Original files for the confocal images presented in *Figure 2*.

**Source data 3.** Numerical values of graphs are shown in *Figure 2*.

**Figure supplement 1.** Compromised organ size due to ubiquitous depletion of *Nup107*: *Actin5C-Gal4* was used as a ubiquitous driver.

**Figure supplement 1—source data 1.** Original confocal images are presented for *Figure 1—figure supplement 1*.

**Figure supplement 2.** Ubiquitous knockdown of *Nup107* using *Nup107^GD* RNA interference (RNAi) disrupts ecdysone signaling.

**Figure supplement 2—source data 1.** Original images for *Figure 2—figure supplement 2* are shown.

**Figure supplement 2—source data 2.** Original files for the confocal images presented in *Figure 2—figure supplement 2*.

**Figure supplement 2—source data 3.** Numerical values of graphs are shown in *Figure 2—figure supplement 2*.

control (*Figure 3—figure supplement 2*). However, the PG-specific knockdown induced an extended third instar stage lifespan (10–12 days AEL, *Figure 3—figure supplement 2*). The observed decrease in ecdysone-inducible gene expression during late third-instar developmental stages can explain the potential impairment of metamorphosis induction seen upon ubiquitous or PG-specific *Nup107*

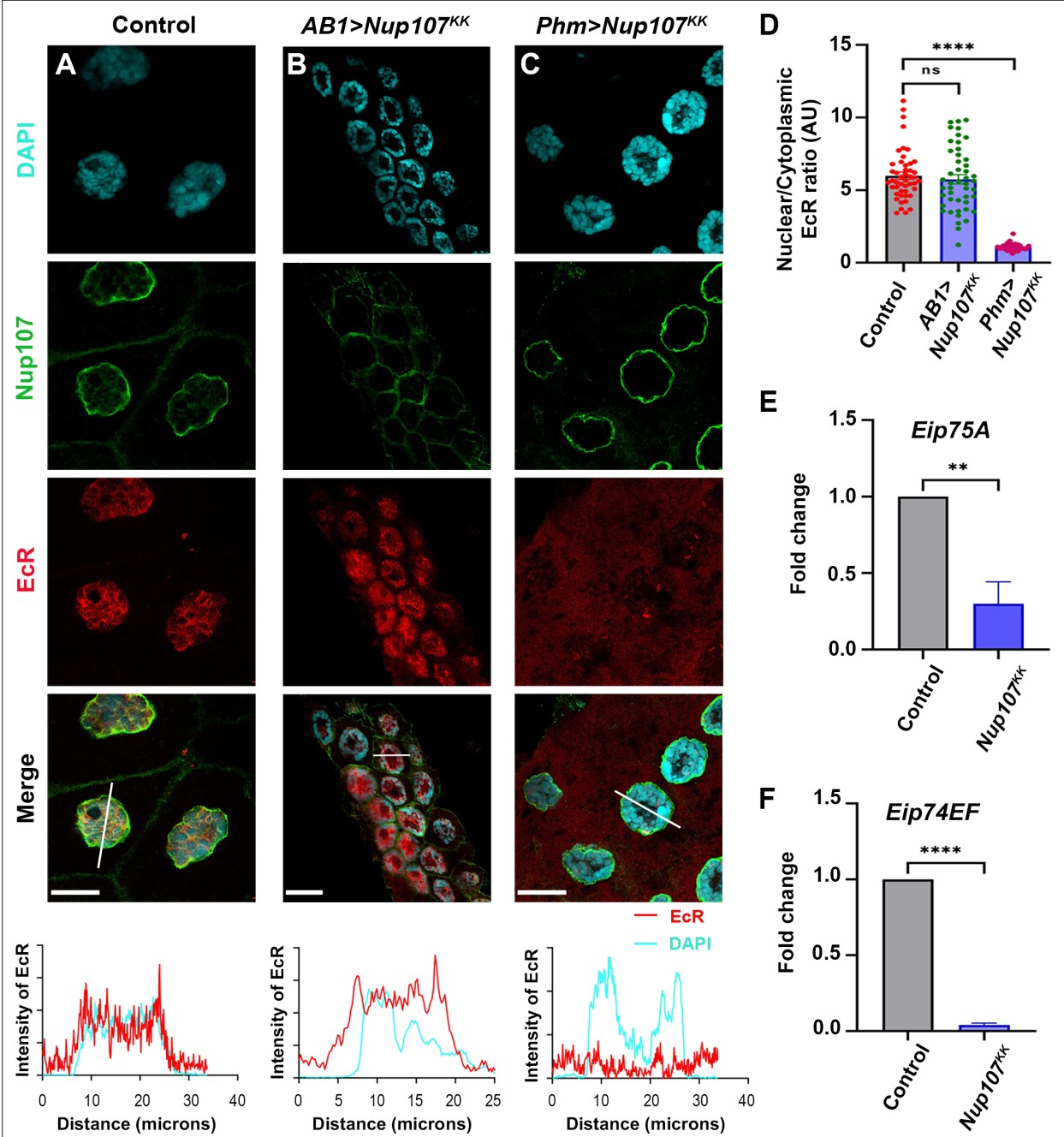

**Figure 3.** Targeted knockdown of *Nup107* in the prothoracic gland (PG) perturbs ecdysone signaling pathway. Analyzing the impact of tissue-specific *Nup107* depletion on ecdysone receptor (EcR) signaling in third instar larval salivary glands. (**A–C**) Detection and quantitation of nucleocytoplasmic distribution of EcR (anti-EcR antibody, red) and Nup107 (anti-Nup107 antibody, green) in control (**A**), salivary gland-specific *Nup107* depletion (**B**), and prothoracic gland-specific *Nup107* depletion (**C**) from third instar larval salivary gland nuclei. DNA is stained with DAPI. Scale bars, 20 µm. Charts represent the line scan intensity profile of EcR (red) and DAPI (cyan) in the salivary gland nucleus region. (**D**) EcR nucleocytoplasmic quantification ratio from the salivary gland and prothoracic gland-specific *Nup107* knockdown, respectively. At least 45 nuclei were analyzed from seven to eight pairs of salivary glands. Statistical significance was derived from the Student's t-test. Error bars represent SEM. ****p=<0.0001, and ns is nonsignificant. (**E–F**) Quantitation of expression of *Eip75A* (**E**) and *Eip74EF* (**F**) ecdysone-inducible genes at the onset of metamorphosis (RNA isolated from late third instar larvae of control and prothoracic gland-specific *Nup107* depletion). Data are represented from at least three independent experiments. Statistical significance was derived from the Student's t-test. Error bars represent SEM. **p=<0.008 and ****p=<0.0001.

The online version of this article includes the following source data and figure supplement(s) for figure 3:

**Source data 1.** Original confocal images for *Figure 3*, showing the EcR localization in Nup107-depleted tissues.

*Figure 3 continued on next page*

*Figure 3 continued*

**Source data 2.** Original files for the confocal images presented in *Figure 3*.

**Source data 3.** Numerical values of graphs are shown in *Figure 3*.

**Figure supplement 1.** *Nup107$^{GD}$* RNA interference (RNAi)-mediated *Nup107* depletion regulates ecdysone receptor (EcR)-dependent signaling.

**Figure supplement 1—source data 1.** Original images for *Figure 3—figure supplement 1* are shown.

**Figure supplement 1—source data 2.** Original files for the confocal images presented in *Figure 3—figure supplement 1*.

**Figure supplement 1—source data 3.** Numerical values of graphs are shown in *Figure 3—figure supplement 1*.

**Figure supplement 2.** *Nup107* regulates metamorphosis via ecdysone synthesis.

**Figure supplement 2—source data 1.** Original images corresponding to *Figure 3* and *Figure 3—figure supplement 2*.

**Figure supplement 2—source data 2.** Original files for *Figure 3—figure supplement 2*.

**Figure supplement 2—source data 3.** Numerical values of graphs are shown in *Figure 3—figure supplement 2*.

**Figure supplement 3.** *Nup107* depletion compromises the prothoracic gland (PG) size.

**Figure supplement 3—source data 1.** The original confocal images of the prothoracic glands correspond to *Figure 3—figure supplement 3*.

**Figure supplement 3—source data 2.** Original files for *Figure 3—figure supplement 3*.

knockdown. In addition to the reduced size of the salivary gland and brain complex, we also noticed a compromise in the size of the PG upon *Nup107* knockdown (*Figure 3—figure supplement 3*).

These observations suggest that Nup107 exerts a regulation on ecdysone biosynthesis and active 20E-EcR complex formation rather than playing a direct role in EcR nuclear translocation.

## *Nup107* exerts control on the EcR pathway through ecdysone level regulation

We reasoned that Nup107 may regulate the ecdysone biosynthesis in PG to induce the larval stage growth arrest. We delved into analyzing the effect of *Nup107* knockdown on ecdysone production. The considerable decrease in PG size due to *Nup107* knockdown (*Figure 3—figure supplement 3*) can potentially reduce 20E production. This prompted us to explore the potential role of Nup107 in influencing ecdysone production, and we assessed the impact of *Nup107* knockdown on ecdysone biosynthesis.

Utilizing an enzyme-linked immunosorbent assay (ELISA)-based detection method, we assessed 20E levels in larvae at 96 and 120 hr AEL. Larvae from different experimental conditions, including control, ubiquitous *Nup107* depletion, and PG-specific *Nup107* depletion, were used in this analysis. Strikingly, the results indicated a substantial decrease in total 20E levels upon Nup107 knockdown (approximately threefold and ninefold, respectively, for ubiquitous and PG-specific knockdown), particularly at 120 AEL, which coincides with the 20E surge seen during metamorphosis (*Figure 4A*, *Figure 4—figure supplement 1*). This observation is crucial and suggests a potential defect in ecdysone biosynthesis in *Nup107*-depleted organisms.

As depicted in *Figure 4B*, the PTTH hormone signaling in the PG upregulates the expression of *Halloween* genes (*spookier*, *phantom*, *disembodied*, and *shadow*) responsible for ecdysone biosynthesis, and the *shade* gene product is required for active 20E generation in peripheral tissues (*Christensen et al., 2020*; *McBrayer et al., 2007*; *Shimell et al., 2018*). We analyzed the *Halloween* genes transcript level in *Nup107*-depleted larvae and observed a significant downregulation for each of the *Halloween* quartet genes mentioned earlier (*Figure 4C–F*, *Figure 4—figure supplement 1*). Further, the level of the *shade* gene was also reduced twofold in *Nup107*-depleted larvae (*Figure 4G*, *Figure 4—figure supplement 1*).

## 20E rescues Nup107-dependent EcR localization defects

The observed correlation between the expression levels of ecdysone biosynthetic genes and reduced levels of 20E upon *Nup107* knockdown strongly suggests a role for Nup107 in 20E biosynthesis, active 20E-EcR complex formation, and subsequent nuclear translocation. It is not surprising in this context that exogenous ecdysone supplementation through larval food rescues pupariation blocks arising from ecdysone deficiency (*Garen et al., 1977*; *Ou et al., 2016*; *Shimell et al., 2018*). We experimented the same under *Nup107* depletion conditions and supplemented exogenous

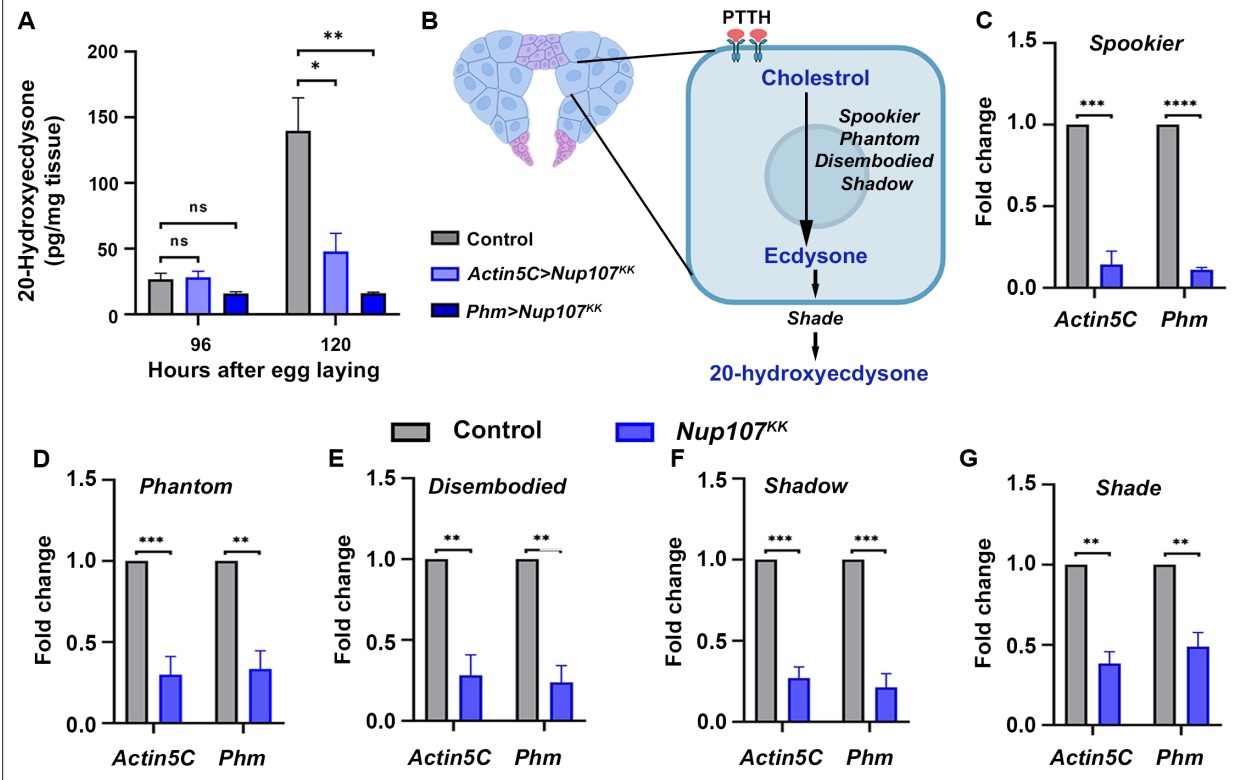

**Figure 4.** Nup107 critically regulates the expression of ecdysone-biosynthetic genes. (**A**) Enzyme-linked immunosorbent assay (ELISA) measurements of whole-body 20-hydroxyecdysone (20E) levels in control, ubiquitous (Actin5C-Gal4), and prothoracic gland-specific (Phm-Gal4) Nup107 depletion at 96 and 120 hr after egg laying (AEL). Data are represented from at least three independent experiments. Statistical significance was derived from the Student's t-test. Error bars represent SEM. *p=<0.032, **p=<0.0078, and ns is nonsignificant. (**B**) Schematic representation of a prothoracic gland cell showing genes involved in ecdysone biosynthesis from cholesterol. (**C–G**) Quantification of ecdysone-biosynthetic gene expression levels of *spookier* (**C**), *phantom* (**D**), *disembodied* (**E**), *shadow* (**F**), and *shade* (**G**) in cDNA isolated from control and Nup107 knockdown late third instar larvae. Data are represented from at least three independent experiments. Statistical significance was derived from the Student's t-test. Error bars represent SEM. **p=<0.001, ***p=<0.0005, and ****p=<0.0001. Created with BioRender.com.

The online version of this article includes the following source data and figure supplement(s) for figure 4:

**Source data 1.** Numerical values of graphs are shown in *Figure 4*.

**Figure supplement 1.** *Nup107^GD* RNA interference (RNAi)-mediated depletion of *Nup107* critically regulates the expression of ecdysone-biosynthetic genes.

**Figure supplement 1—source data 1.** Numerical values of graphs are shown in *Figure 4—figure supplement 1*.

ecdysone to PG-specific *Nup107*-depleted larvae by feeding them a diet enriched with 20E (0.2 mg/ml). Supplementation of 20E to *Nup107*-depleted larvae significantly alleviated the developmental arrest, and the onset of pupariation was comparable to the control (*Supplementary file 1*). However, none of the pupae could eventually eclose successfully, probably due to other effects of *Nup107* depletion. We asked if nuclear translocation of EcR can also be rescued by exogenous supplementation of 20E. While incubation of salivary glands of late third instar larva in S2 media control alone did not rescue EcR localization in any of the *Nup107* knockdown genotypes (*Figure 5A–C*, *Figure 5—figure supplement 1*), we noticed a complete EcR nuclear translocation rescue in ubiquitous, as well as PG-specific *Nup107*-depleted salivary glands when incubated with 20E (*Figure 5D–F*, *Figure 5—figure supplement 1*).

We reason that the exogenous supplementation of 20E leads to active 20E-EcR complex formation as mRNA levels of the *Eip75A* and *Eip74EF* target genes were rescued significantly back to normal levels (*Figure 5G and H*). Overall, these findings highlight the important regulatory function of Nup107 in the ecdysone signaling pathway.

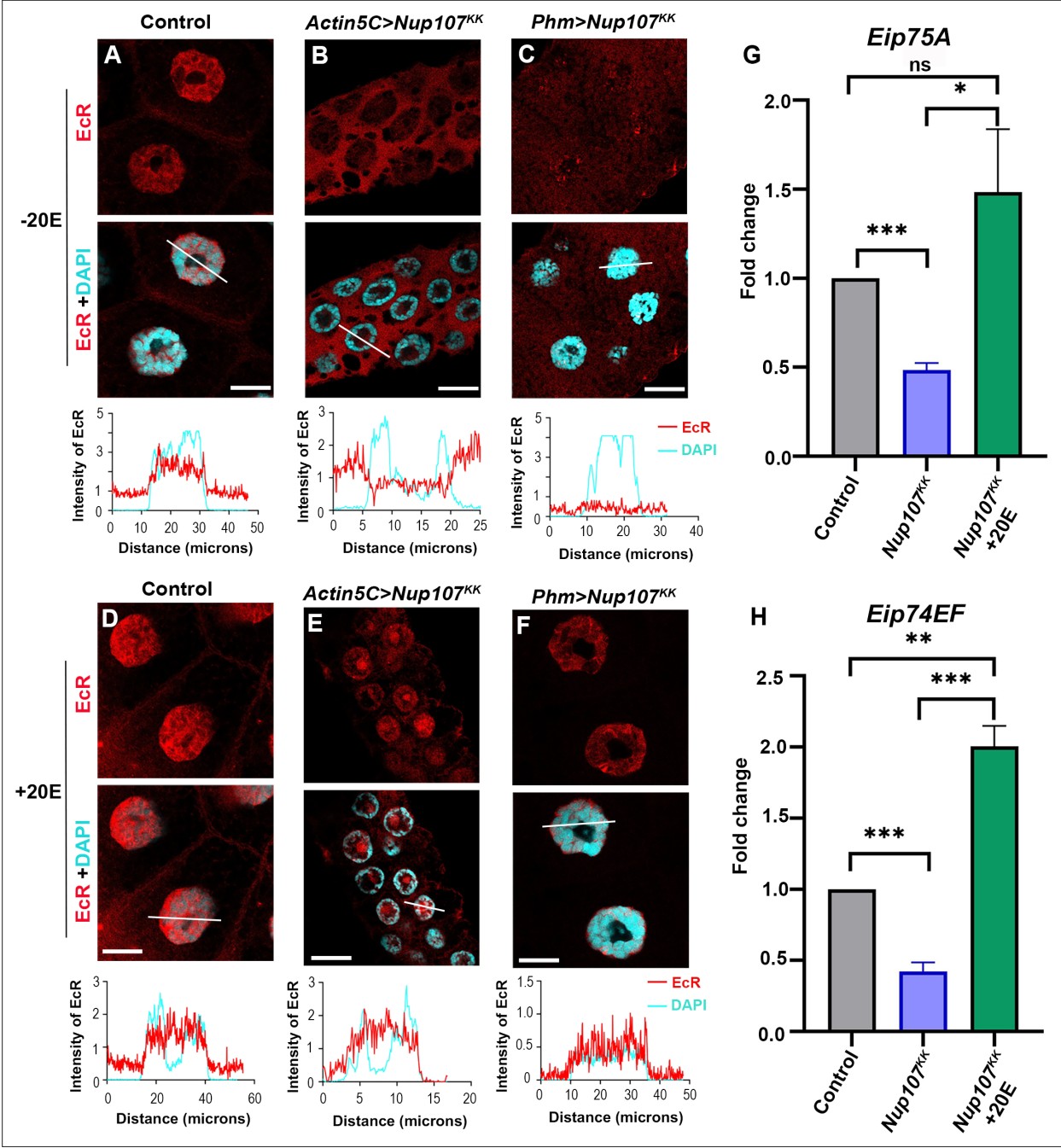

**Figure 5.** 20-Hydroxyecdysone (20E) supplementation rescues Nup107 depletion-specific ecdysone receptor (EcR) signaling defects. Immunofluorescence analysis of EcR localization in 20E supplemented and non-supplemented third instar larval salivary glands. (**A–F**) Visualization of the nucleocytoplasmic distribution of EcR (anti-EcR antibody, red) without 20E (**A–C**) and with 20E (**D–F**) treatment in larval salivary glands of control, ubiquitous (Actin5C-Gal4) Nup107 knockdown, and prothoracic gland-specific (Phm-Gal4) Nup107 knockdown. DNA is stained with DAPI. Scale bars, 20 μm. Charts represent the line scan intensity profile of EcR (red) and DAPI (cyan) in the salivary gland nucleus region. (**G–H**) Comparative quantification of expression ecdysone-inducible genes Eip75A (**G**) and Eip74EF (**H**) from 20E-treated salivary glands. Data are represented from at least three independent experiments. Statistical significance was derived from the Student's t-test. Error bars represent SEM. *p=<0.02, **p=<0.002, ***p=<0.0008, and ns is nonsignificant.

The online version of this article includes the following source data and figure supplement(s) for figure 5:

**Source data 1.** Original confocal images for *Figure 5*, showing the EcR localization with and without 20E.

**Source data 2.** Original files for *Figure 5*.

**Source data 3.** Numerical values of graphs are shown in *Figure 5*.

*Figure 5 continued on next page*

*Figure 5 continued*

**Figure supplement 1.** Ecdysone (20-hydroxyecdysone [20E]) supplementation rescues *Nup107^GD*-dependent *Nup107* depletion-specific ecdysone receptor (EcR) signaling defects.

**Figure supplement 1—source data 1.** Original images for without 20-hydroxyecdysone (20E) (*Figure 5—figure supplement 1A*) and with 20E (*Figure 5—figure supplement 1B*) are shown.

**Figure supplement 1—source data 2.** Original files for *Figure 5—figure supplement 1*.

**Figure supplement 1—source data 3.** Numerical values of graphs are shown in *Figure 5—figure supplement 1*.

## *Torso* is an effector of *Nup107*-mediated functions in metamorphosis

Next, we explored the mechanism of Nup107-driven regulation of 20E levels and metamorphosis. The cell surface receptors of the tyrosine kinase family bind to ligands (growth factors and hormones) and activate signaling to regulate metabolism, cell growth, and development. Among these RTKs, the Torso, belonging to the platelet-derived growth factor receptor class, plays a significant role during metamorphosis by serving as a receptor for the neuropeptide PTTH in the *Drosophila* brain (*Sopko and Perrimon, 2013*).

Functional engagement of PTTH-Torso activates the MAP kinase pathway, involving components of Ras, Raf, MEK, and ERK, thereby initiating metamorphosis (*Figure 6A*). The *torso* knockdown in the PGs resulted in a significant delay in the onset of pupariation, extending the developmental period by approximately 6 days, resembling the developmental arrest seen with *Nup107* knockdown. Feeding 20E to *torso*-depleted larvae completely rescued developmental delay and normal growth phenotypes (*Rewitz et al., 2009*). We observed a significant decrease in the *torso* levels (approximately fourfold) when *Nup107* was depleted ubiquitously using *Actin5C-GAL4* (*Figure 6B*), suggesting an epistatic regulation by Nup107 on the torso. Further, the phenotypic similarity between *Nup107* and *torso* depletion scenarios and diminished *torso* level upon *Nup107* depletion prompted us to investigate whether the *torso* is an effector of Nup107.

Overexpression of the *torso* and/or *ras^V12* has been utilized to rescue torso pathway-mediated defects (*Cruz et al., 2020*; *Rewitz et al., 2009*). We probed the possibility of *Nup107* phenotype rescue by ubiquitous or PG-specific overexpression of the *torso* and *ras^V12*. Overexpression of either *torso* or *ras^V12* in *Nup107* depletion background completely rescued the pupariation defects (*Figure 6C and D*). Interestingly, the overexpression of *Egfr* (another RTK linked to *Drosophila* metamorphosis) and *Usp* (co-receptor with EcR) in *Nup107* depletion background could not rescue the pupariation defects (data not shown), indicating that the *torso* overexpression-mediated rescue of Nup107 depletion phenotypes is specific. Overexpression-dependent rescue prompted us to analyze the status of EcR nuclear translocation, ecdysone biosynthesis, and ecdysone-inducible gene expression. The restoration of EcR nuclear translocation (*Figure 6E–G*), the ecdysone biosynthesis genes, *spookier*, *phantom*, *disembodied*, and *shadow* (*Figure 6—figure supplement 1*), and ecdysone target genes *Eip75A* and *Eip74EF* (*Figure 6H and I*) to control levels indicated that *torso* could efficiently rescue *Nup107* phenotypes.

These observations suggest that the *torso* is a downstream effector of Nup107 functions, and torso-dependent signaling, responsible for 20E synthesis in PG and metamorphosis, is regulated by Nup107 levels.

## Discussion

Apart from its importance in mRNA export and cell division at the cellular level, the NPCs Y-complex member Nup107, when mutated, correlates with developmental abnormalities such as microcephaly and nephrotic syndrome (*Miyake et al., 2015*; *Zheng et al., 2012*). In several human cancer types, Nup107 exhibits high expression and serves as a biomarker for hepatocellular carcinoma. A strong structural and functional conservation between human and *Drosophila* Nup107 proteins can help us model human diseases and gain mechanistic insights in Nup107 functions in *Drosophila* (*Shore et al., 2022*; *Weinberg-Shukron et al., 2015*). We observe that Nup107 is involved in critical developmental transitions from larva to pupa during *Drosophila* development as pupariation is completely arrested (*Figure 1*). The foraging third instar larva must acquire a critical weight and sufficient energy stores before it can metamorphose into a non-feeding pupa, molting into a healthy adult. The activation

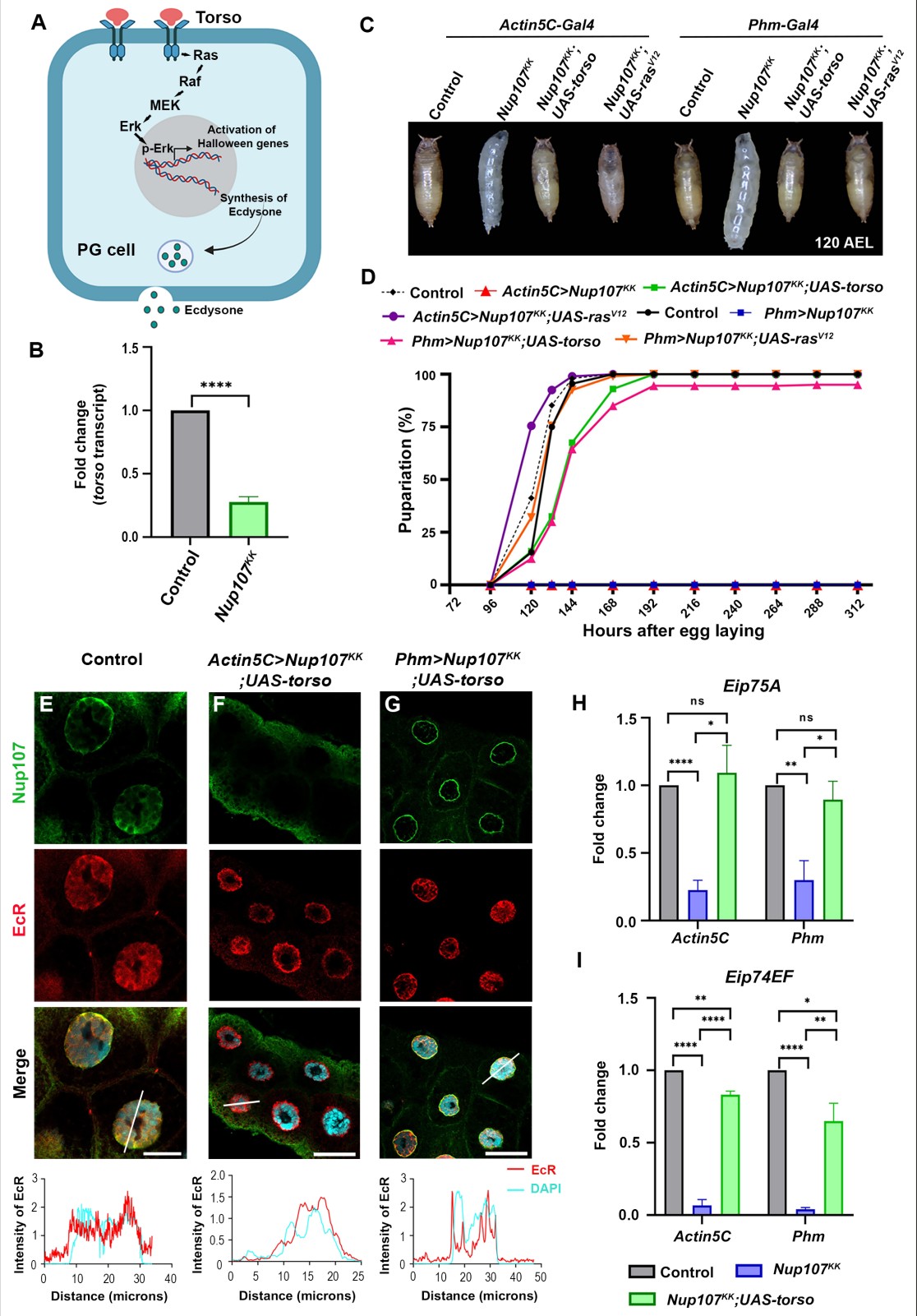

**Figure 6.** Torso and Nup107 act synergistically to activate the ecdysone signaling. (**A**) A model of the Torso pathway and its components. (**B**) Quantitation of torso transcript levels from control and Nup107-depleted larvae (ubiquitous depletion using Actin5C-Gal4). Data are represented from at least three independent experiments. Statistical significance was derived from the Student's t-test. Error bars represent SEM. ****p=<0.0001. (**C–D**) Comparison of pupariation profiles of control, Nup107 knockdown, and torso and rasV12 overexpressing rescue organisms. (**E–G**) Detection and

*Figure 6 continued on next page*

*Figure 6 continued*

quantitation of nucleocytoplasmic distribution of EcR (anti-EcR antibody, red) and Nup107 (anti-Nup107 antibody, green) in control, torso overexpressing ubiquitous Nup107 knockdown (Actin5C-Gal4>Nup107KK; UAS-torso) and torso overexpressing PG-specific Nup107 knockdown (Phm-Gal4>Nup107KK; UAS-torso) third instar larval salivary gland nuclei. DNA is stained with DAPI. Scale bars, 20 µm. Charts show the line scan intensity profiles of EcR (red) and DAPI (cyan) in the salivary gland nucleus region. (**H–I**) Quantification of expression of Eip75A (**H**) and Eip74EF (**I**) ecdysone-inducible genes, respectively. Data are represented from at least three independent experiments. Statistical significance was derived from the Student's t-test. Error bars represent SEM. *p=<0.03, **p=<0.008, ****p=<0.0001, and ns is nonsignificant. Created with BioRender.com.

The online version of this article includes the following source data and figure supplement(s) for figure 6:

**Source data 1.** Original confocal images for the confocal images presented in *Figure 6*.

**Source data 2.** Original files for the confocal images presented in *Figure 6*.

**Source data 3.** Numerical values of graphs are shown in *Figure 6*.

**Figure supplement 1.** *Torso* rescues metamorphic defects of *Nup107* knockdown.

**Figure supplement 1—source data 1.** Uncropped image of larvae corresponding to *Figure 6—figure supplement 1*.

**Figure supplement 1—source data 2.** Original files of larval images of *Figure 6—figure supplement 1*.

**Figure supplement 1—source data 3.** Numerical values of graphs are shown in *Figure 6—figure supplement 1*.

of the Torso receptor and sequent surge in ecdysone synthesis are essential for larval-to-pupal transition (*Christensen et al., 2020*; *Hao et al., 2021*; *Luo et al., 2024*). The elevated ecdysone levels trigger the translocation of the heterodimeric EcR (in complex with ultraspiracle [Usp]) into the nucleus to induce pupariation-specific transcriptional programming (*Johnston et al., 2011*). Similarly, when *Nup107* is depleted, nuclear translocation of EcR is abolished, arresting metamorphosis at the level of the third instar larval stage (*Figure 2*).

Translocation of nuclear receptor proteins like the EcR to the nucleus is a crucial step for transcriptional activation. The marked absence of EcR from the nucleus was intriguing as Nup107 generally participates in the nuclear export process. Perhaps Nup107 regulates pupariation by modulating ecdysone synthesis, ecdysone signaling, and nuclear translocation of EcR. Surprisingly, Nup107 is dispensable for the nuclear translocation of EcR in the target tissue, the salivary glands (*Figure 3*). Nup107 exerts strong control over the expression of *Halloween* genes involved in ecdysone biosynthesis, resulting in diminished 20E titer, poor EcR activation, and delayed larva-to-pupa transition (*Figure 4*, *Figure 3—figure supplement 2*). The effect of Nup107 on *Halloween* genes adheres well to pupariation defects. These observations are in concurrence with reports of low ecdysone levels disrupting the pupariation process, leading to a halt in insect development (*Christensen et al., 2020*; *Cruz et al., 2020*).

In addition to ecdysone signaling, insect metamorphosis also has a strong contribution from RTK pathway signaling. Notably, Torso signaling is essential for embryonic development and metamorphosis in *Drosophila*. The receptor expression level impacts the signaling output: high receptor levels trigger a robust and transient signal, while lower levels result in a weaker, sustained signal (*Jenni et al., 2015*; *Konogami et al., 2016*). Accordingly, the reduced ecdysone levels in the *torso* knockdown PGs induced pupariation delays (*Rewitz et al., 2009*). We observe a similar defect with *Nup107* knockdown, indicating a possible reduction in *torso* levels. Interestingly, we observed significantly reduced torso levels in *Nup107*-depleted organisms, suggesting an essential upstream function for Nup107 in regulating RTK pathway activation and the associated *Drosophila* metamorphosis process. In the contrasting observation, *Nup107*-depleted larval tissues (salivary gland, brain complex, and PG) are significantly smaller in size (*Figure 2—figure supplement 1*, *Figure 3—figure supplement 3*). It is important to know how Nup107 contributes to organ size maintenance and if a crosstalk with RTK signaling is required in this context.

The developmental delays observed in *Nup107* knockdown larvae can be restored by exogenous supplementation of 20E, suggesting that Nup107 can modulate directly or indirectly the Torso receptor activity for ecdysone production and developmental regulation. Accordingly, we successfully rescued developmental delays by ubiquitous and PG-specific *torso* overexpression. Perhaps overexpression of the torso pathway mediator molecules can rescue the *Nup107* developmental delay phenotypes. The oncogenic variant of the Ras-GTPase ($ras^{V12}$) has been explored in a similar analysis with torso pathway mutants (*Cruz et al., 2020*; *Rewitz et al., 2009*). The alleviation of the developmental delays in the

*Nup107* knockdown background upon *ras*$^{V12}$ overexpression is an indication of Nup107 serving as an epistatic regulator of the torso pathway in metamorphic transitions (*Figure 6*).

In essence, our findings indicate that Nup107 influences pupariation timing by regulating the torso levels, its signaling, and ecdysone biosynthesis (*Figure 7*). Previous research has shown that NPCs are essential for maintaining global genome organization and regulating gene expression (*Capelson et al., 2010a*; *Capelson et al., 2010b*; *Iglesias et al., 2020*). Particularly, Nup107 interacts with chromatin and targets active gene domains to regulate gene expression (*Gozalo et al., 2020*). Thus, Nup107 exerting its effects on *torso* transcription is the primary regulatory event in the *Drosophila* metamorphosis. It is important to note that the synthesis and availability of Torso ligand, PTTH, may not be affected by the Nup107 since torso or downstream effector *ras*$^{V12}$ overexpression is sufficient to rescue the developmental arrest phenotype. It is thus crucial to further delineate the mechanism of Nup107-dependent regulation on *torso* pathway activation. The whole-genome transcriptomics from PG can help shed more light on the regulatory roles of Nup107. This information will offer valuable insights into how Nups regulate gene expression and serve as a model for elucidating how they govern the temporal specificity of developmental processes in organisms. Furthermore, these analyses will help establish a link between Nup107, the PTTH-PG axis, and the regulation of developmental transition timing. Our observations indicate critical roles for Nup107 in both torso-ecdysone interplay in metamorphosis and torso-independent mechanism for organ size maintenance. Together they add valuable information to hitherto unknown functions of the Nup107 in organismal development. Further investigations are required to identify the interactors of Nup107 involved in these coordination mechanisms in developmental transition.

## Materials and methods
### Fly stocks and genetics
Experimental *Drosophila melanogaster* stocks were reared on a standard cornmeal diet (Nutri-Fly Bloomington formulation) under controlled conditions of 25°C and 60% relative humidity unless otherwise specified. Fly lines used in this study were sourced from the Bloomington Drosophila Stock Centre (BDSC) at Indiana University or from the Vienna Drosophila Resource Center (VDRC), which are listed in the Key resources table. The UAS-GFP lines were received as a gift from Dr. Varun Chaudhary (IISER Bhopal, India). Control groups were generated through crosses between the driver line and *w*$^{1118}$ flies. For RNAi experiments, crosses were maintained at 29°C to optimize GAL4 expression. The *Nup107*$^{GD}$ line, in conjunction with *Actin5C-GAL4*, was specifically cultivated at 23°C to generate third instar larvae for experimental purposes. Various genetic combinations were generated as per the need of the experiment following the standard *Drosophila* genetics cross schemes.

### Transgenic fly generation
In the generation of transgenic flies containing gRNA, we employed a systematic approach. The online tool available at http://targetfinder.flycrispr.neuro.brown.edu was used for the initial sgRNA design, ensuring zero predicted off-targets. Subsequently, we utilized an additional tool available at https://www.flyrnai.org/evaluateCrispr/ to evaluate and score the predicted efficiency of sgRNAs in the targeted region. The most efficient sgRNAs, demonstrating high specificity with no predicted off-targets, were selected and cloned into the pCFD4 vector. The Fly Facility Services at the Centre for Cellular and Molecular Platforms, National Center for Biological Sciences (C-CAMP-NCBS), Bengaluru, India, were utilized to clone and generate transgenic flies. Primer sequences employed in this study can be found in the Key resources table.

### CRISPR-Cas9-mediated mutant generation
Virgin *nanos.Cas9* (BL-54591) flies were crossed with males of gRNA transgenic lines, and approximately 10 F1 progeny males were crossed with balancer flies specific for the gene of interest. A single-line cross was established using the F2 progeny to assess the deletion of *nup107*. Subsequently, F3 progeny were subjected to nested PCR to confirm deletions. Positive individuals were further tested for the presence of gRNA and Cas9 transgenes, and flies lacking both were propagated into stocks.

### Genomic DNA isolation
Approximately 10 flies were homogenized in 250 µL of solution A, comprising 0.1 M Tris-HCl, pH 9.0, 0.1 M EDTA, and 1% SDS, supplemented with Proteinase K. The homogenate was incubated at 70°C

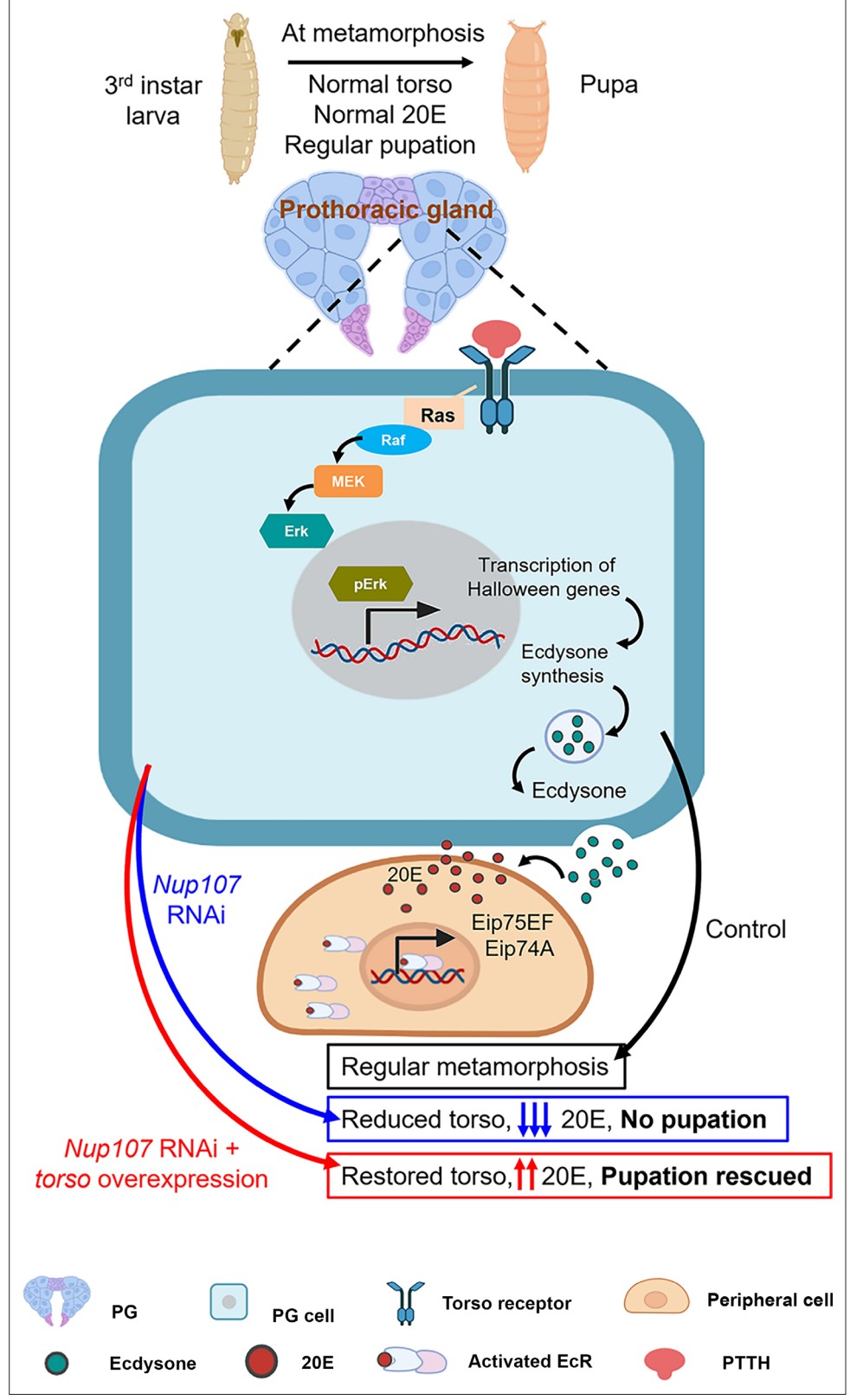

**Figure 7.** Theoretical model of Nup107 functions in metamorphosis. During metamorphosis, the prothoracic gland (PG) responds to prothoracicotropic hormone (PTTH) via Torso receptors. The MAP kinase pathway involving Raf, MEK, and ERK, initiated by Ras, leads to *Halloween* gene expression responsible for ecdysone synthesis and release. Ecdysone is converted to active 20-hydroxyecdysone (20E) in peripheral tissues. The binding of 20E to

*Figure 7 continued on next page*

*Figure 7 continued*

the ecdysone receptor (EcR) allows EcR nuclear translocation and EcR pathway activation, culminating in target gene expression facilitating the metamorphic transition. Nup107 depletion negatively impacts Torso levels and Torso pathway activation, inducing pupariation arrest, which can be rescued by autonomous activation of the Torso pathway. Created with BioRender.com.

for 30 min. Subsequently, 35 µL of 8 M potassium acetate was added, mixed gently without vortexing, and incubated on ice for 30 min. The homogenate was then centrifuged at 13,000 rpm at 4°C for 15 min, and the supernatant was carefully collected without disturbing precipitates or the interphase. 150 µL of isopropanol was added to the supernatant and incubated on ice for 15 min. After centrifugation at 13,000 rpm for 5 min, the supernatant was discarded, and the pellet was washed with 1 mL of 70% ethanol by centrifuging at 13,000 rpm for 5 min. The supernatant was again discarded, and the pellet was dried at 55°C until the ethanol evaporated. Finally, the pellet was resuspended in 50 µL of Tris-EDTA buffer and incubated at 37°C.

## Nested PCR

Nested PCR was employed to assess the presence of deletions in flies, utilizing genomic DNA as the template. 1 µL of DNA was used for the initial PCR with an outer set of primers (Key resources table). Following this, 1 µL of the initial PCR product was employed for a subsequent PCR with an inner set of primers. The resulting PCR products were then resolved on 0.8% agarose gel to visualize the desired bands indicative of deletions.

## Measurement of developmental timing and pupariation

Flies were permitted to lay eggs for 3 hr on agar plates supplemented with yeast, synchronizing the larvae. Newly hatched L1 larvae were then collected and transferred to vials. Pupariation times and dates were recorded daily during the light cycle. Data from 8 to 10 vials were aggregated, organized by pupariation time and cumulative percentage pupariation, and analyzed using Microsoft Excel.

## Antibody generation and western blotting

To generate antibodies against *Drosophila Nup107* (CG6743), the N-terminal 210 amino acids, recognized as the most unique and antigenic region, were sub-cloned into the modified tag-less pET28a(+) vector, pET28a(+)-JK vector. The protein was expressed in *Escherichia coli* BL21 (DE3) cells, induced with 200 µM IPTG (Sigma), and incubated at 30°C for 4 hr. Following cell pelleting, the pellet was resuspended in lysis buffer (50 mM Tris pH 8.0, 1 mM EDTA, and 25% sucrose) with 1× protease inhibitor mixture (Roche Applied Science), and lysozyme (from 50 mg/mL stock) was added to facilitate bacterial lysis. After sonication and centrifugation, the pellet was successively resuspended in inclusion body buffer I (20 mM Tris pH 8.0, 0.2 M NaCl, 1% sodium deoxycholate, and 2 mM EGTA) and buffer II (10 mM Tris pH 8.0, 0.25% sodium deoxycholate, and 1 mM EGTA) and centrifuged. This process was repeated three times within inclusion body buffer II, and the final pellet was dissolved in 8 M urea buffer (10 mM Tris-HCl pH 8.0, 8 M urea, 0.1 mM NaN$_3$, and 1 mM EGTA), diluted to 6 M urea, and centrifuged. The supernatant was loaded onto SDS-PAGE, and the desired band was cut for protein elution. Rabbit polyclonal antibodies against Nup107 were generated at Bio Bharati Life Sciences Pvt. Ltd., Kolkata, West Bengal, India. The antibodies obtained were subjected to affinity purification. The purification involved the chemical cross-linking of purified antigens to N-hydroxy succinimidyl-Sepharose beads (Sigma). The elution process was carried out under low pH conditions, followed by neutralization. Subsequently, the eluted antibodies were dialyzed against phosphate-buffered saline (PBS) overnight at 4°C to remove any remaining impurities and optimize their stability and functionality for subsequent experimental use.

Larval brain complexes from third instar larvae were dissected, lysed in Laemmli buffer, and resolved on 8% SDS-PAGE. Two equivalent head complexes were loaded per well. After transfer to a polyvinylidene difluoride membrane, blocking was performed with 5% fat-free milk. The membrane was then incubated with polyclonal anti-Nup107 antibody (1:500) and anti-α-tubulin (DSHB, 12G10) (1:5000) at 4°C overnight. Following three washes with TBS-T buffer (20 mM Tris-HCl pH 7.5, 150 mM NaCl, 0.1% Tween-20), the membrane was incubated with secondary antibodies, anti-rabbit-Alexa Fluor Plus 680 (Thermo Fisher Scientific #A32734) and anti-mouse-Alexa Fluor Plus 800 (Thermo Fisher Scientific

#A32730). Following incubation with secondary antibodies, the membrane was washed three times for 10 min each with Tris-buffered saline containing Tween-20 (TBS-T) to remove unbound antibodies. The washed membrane was then subjected to imaging using the Li-COR IR system (Model: 9120), allowing for the visualization and analysis of the protein bands on the membrane.

## Immunostaining

For immunostaining *Drosophila* salivary glands, the previously reported protocol was followed (*Mehta et al., 2021*). The larvae were dissected in cold PBS to isolate salivary glands. Glands underwent pre-extraction with 1% PBST (PBS+1% Triton X-100), then fixation in freshly prepared 4% paraformaldehyde for 30 min at room temperature. Subsequently, the glands were thoroughly washed with 0.3% PBST (0.3% Triton X-100 containing PBS). Blocking was performed for 1 hr with 5% normal goat serum (#005-000-001, The Jackson Laboratory, USA). The glands were stained with the following primary antibodies: anti-Nup107 (1:100), mAb414 (1:500, BioLegend), and anti-EcR (1:20, DDA2.7, DSHB) overnight at 4°C. Tissues were washed three times with 0.3% PBST (PBS+0.3% Triton X-100), followed by incubation with secondary antibodies: anti-rabbit Alexa Fluor 488 (1:800, #A11034, Thermo Fisher Scientific), anti-rabbit Alexa Fluor 568 (1:800, #A11036, Thermo Fisher Scientific), anti-mouse Alexa Fluor 488 (1:800, #A11029, Thermo Fisher Scientific), anti-mouse Alexa Fluor 568 (1:800, #A11004, Thermo Fisher Scientific), and anti-rabbit Alexa Fluor 647 (1:800, Jackson ImmunoResearch). Following secondary antibody incubation, tissues were washed three times with 0.3% PBST (PBS+0.2% Triton X-100) and mounted with DAPI-containing Fluoroshield (#F6057, Sigma). The same protocol was followed for staining of the brain complex and PG.

## Fluorescence intensity quantification

Images were acquired using an Olympus Confocal Laser Scanning Microscope FV3000. Subsequent image processing was conducted using the Fiji software, and signal intensities were averaged using GraphPad software (Prism). To quantify the EcR nuclear-to-cytoplasmic ratio, three distinct regions of interest (ROIs) were designated per nucleus and its surrounding cytoplasm. Fiji software measured the average intensity in each ROI, and the mean of these intensities was determined per nucleus and its adjacent cytoplasm. The final graph depicts the ratio of the mean intensities observed in the nucleus to that in the cytoplasm. All experiments were conducted with a minimum of three independent replicates.

## Quantitative RT-PCR

Total RNA was extracted from various genotypes (control, ubiquitously depleted *Nup107*, and PG-specific depletion of *Nup107*) of whole late third instar larvae utilizing the total tissue RNA isolation kit (Favorgen Biotech). One microgram of total RNA was employed for cDNA synthesis with the iScript cDNA synthesis kit (Bio-Rad). The resulting cDNA was diluted fivefold, and 1 µL from each genotype was utilized as a template. RT-PCR was conducted using SYBR Green PCR master mix on an Applied Biosystems QuantStudio 3 Real-Time PCR System. The Rpl69 gene served as the control, and relative transcript levels were determined using the CT value ($2^{-\Delta\Delta CT}$). Differences in gene expression were analyzed using Student's t-test, with a p-value<0.05 considered significant. The primers used are detailed in the Key resources table. The resultant graph illustrated the fold change in gene expression, plotted using GraphPad software (Prism).

## 20E level measurements

Three biological replicates of 25 mg larvae from each genotype were collected at the specified times AEL. The larvae were washed, dried, and weighed before being flash-frozen on dry ice and stored at –80°C. Ecdysone extraction was done by thoroughly homogenizing the frozen samples in 300 µL ice-cold methanol with a plastic pestle. After centrifugation at 17,000 × *g* for 10 min, the supernatant was divided into two Eppendorf tubes containing approximately 150 µL supernatant. Methanol from both tubes was evaporated in a vacuum centrifuge for 60 min, and the resulting pellets were redissolved by adding 200 µL enzyme immunoassay (EIA) buffer to one of the tubes. After vortexing, the same 200 µL EIA buffer was transferred to the second tube, followed by another round of vortexing. The ELISA was performed using a 20E ELISA kit (#EIAHYD) from Thermo Fisher Scientific.

## 20E rescue experiment

20E (#H5142, Sigma) was dissolved in ethanol to achieve a 5 mg/mL concentration. Salivary glands from third instar larvae of different genotypes were extracted and then incubated for 6 hr in Schneider's S2 media with 50 µM 20E. After the incubation, the glands underwent procedures for immunostaining and RNA extraction.

To facilitate the rescue of the larvae through feeding, 30 third instar larvae, *Nup107*-depleted larvae cultured at 29°C incubator, were first washed with water. They were then transferred to vials containing either 20E (at a final concentration of 0.2 mg/mL) or 95% ethanol (in the same amount as the 20E). Once the larvae were added to the vials, these were returned to the 29°C incubator and observed for pupariation, recording the time of this developmental stage.

## Acknowledgements

We thank the Bloomington Drosophila Stock Centre (BDSC) and Vienna Drosophila Resource Centre (VDRC) for generously providing the fly lines crucial for this study. Special recognition is given to Bio Bharati Life Sciences Pvt. Ltd., Kolkata, India, for their significant contribution to antibody generation and the Developmental Studies Hybridoma Bank (DSHB) for supplying antibodies. Our sincere thanks go to C-CAMP Bengaluru for their instrumental role in generating the transgenic fly. BioRender. com was used to create models wherever necessary. The Indian Institute of Science Education and Research Bhopal Central Instrumentation Facility is acknowledged for its valuable support in DNA sequencing and facilitating access to confocal microscopes. This work is supported by the Science and Engineering Research Board grant no. CRG/2020/000496 provided to RKM.

## Additional information

### Funding

| Funder | Grant reference number | Author |
|---|---|---|
| Science and Engineering Research Board | | Ram Kumar Mishra |
| Indian Council of Medical Research | | Ram Kumar Mishra |
| Indian Institute of Science Education and Research, Bhopal | | Ram Kumar Mishra |
| University Grants Commission | | Jyotsna Kawadkar |
| Science and Engineering Research Board | CRG/2020/000496 | Ram Kumar Mishra |

The funders had no role in study design, data collection and interpretation, or the decision to submit the work for publication.

### Author contributions

Jyotsna Kawadkar, Conceptualization, Formal analysis, Validation, Investigation, Visualization, Methodology, Writing – original draft, Writing – review and editing; Pradyumna Ajit Joshi, Conceptualization, Investigation; Ram Kumar Mishra, Conceptualization, Resources, Data curation, Formal analysis, Supervision, Funding acquisition, Investigation, Methodology, Writing – original draft, Project administration, Writing – review and editing

### Author ORCIDs

Jyotsna Kawadkar https://orcid.org/0009-0009-1483-6507
Ram Kumar Mishra https://orcid.org/0000-0002-9528-0476

Reviewer #1 (Public review): https://doi.org/10.7554/eLife.105165.3.sa1

Reviewer #2 (Public review): https://doi.org/10.7554/eLife.105165.3.sa2
Reviewer #3 (Public review): https://doi.org/10.7554/eLife.105165.3.sa3
Author response https://doi.org/10.7554/eLife.105165.3.sa4

## Additional files

### Supplementary files
Supplementary file 1. Exogenous 20-hydroxyecdysone (20E) supplementation analysis.
MDAR checklist

### Data availability
All relevant data and resources can be found within the article and its supplementary information.

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

# Appendix 1

## Appendix 1—key resources table

| Reagent type (species) or resource | Designation | Source or reference | Identifiers | Additional information |
|---|---|---|---|---|
| Genetic reagent (*D. melanogaster*) | w[1118] | Bloomington Drosophila Stock Center | BDSC:3605; FLYB: FBst000360; RRID:BDSC_3605 | w[1,118] |
| Genetic reagent (*D. melanogaster*) | w1118; P{GD12024}v22407 | Vienna Drosophila Resource Center | VDRC: v22407; FLYB: FBst0454535; RRID:FlyBase_ FBst0454535 | P{GD12024}v22407 (Nup107[GD] RNAi) |
| Genetic reagent (*D. melanogaster*) | P{KK108047}VIE-260B | | VDRC: v110759; FLYB: FBst0482324; RRID:FlyBase_FBst0482324 | P{KK108047}VIE-260B (Nup107[KK] RNAi) |
| Genetic reagent (*D. melanogaster*) | y[1] w[*]; P{w[+mW.hs]=GawB}AB1 | Bloomington Drosophila Stock Center | BDSC:1824; FLYB: FBti0001249; RRID:BDSC_1824 | AB1-Gal4 |
| Genetic reagent (*D. melanogaster*) | y[1] w[*]; P{w[+mC]=phtm-GAL4.O}22 | | BDSC:80577; FLYB: FBti0201787; RRID:BDSC_80577 | Phm-Gal4 |
| Genetic reagent (*D. melanogaster*) | y[1] w[*]; P{Act5C-GAL4-w}E1/CyO | | BDSC:25374; FLYB: FBti0127834; RRID:BDSC_25374 | Actin5C-Gal4 |
| Genetic reagent (*D. melanogaster*) | w[*]; wg[Sp-1]/CyO; P{w[+mC]=mRFP-Nup107.K}7.1 | | BDSC:35517; FLYB: FBti0130064; RRID:BDSC_35517 | mRFP-Nup107 |
| Genetic reagent (*D. melanogaster*) | y[1] M{w[+mC]=nanos-Cas9.P}ZH-2A w[*] | Bloomington Drosophila Stock Center | BDSC:54591; FLYB: FBti0159183; RRID:BDSC_54591 | Nanos.Cas9 |
| Genetic reagent (*D. melanogaster*) | w[*]; TI{w[+mW.hs]=TI}trk[Delta]/CyO; P{w[+mC]=UASp-tor.G}5.7 | | BDSC:92604; FLYB: FBti0216047; RRID:BDSC_92604 | UAS-torso |
| Genetic reagent (*D. melanogaster*) | w[1118]; P{w[+mC]=UAS-Ras85D.V12}TL1 | | BDSC:4847; FLYB: FBti0012505; RRID:BDSC_4847 | UAS-ras[V12] |
| Genetic reagent (*D. melanogaster*) | Nup107[KK];UAS-GFP | | | Generated in the RKM (corresponding authors) laboratory and details are reported in the Materials and methods under Fly stocks and genetics section. |
| Genetic reagent (*D. melanogaster*) | UAS-GFP;Nup107[GD] | | | |
| Genetic reagent (*D. melanogaster*) | UAS-GFP;Phm-Gal4/TM6.Tb | | | |
| Genetic reagent (*D. melanogaster*) | Nup107 gRNA line | | | |
| Genetic reagent (*D. melanogaster*) | Nup107[KK];UAS-torso | | | |
| Genetic reagent (*D. melanogaster*) | Nup107[KK];UAS-rasV12 | | | |
| Antibody | Anti-Nup107 (Rabbit polyclonal) | This study | | WB (1:500), IF (1:100) Generated in the RKM (corresponding authors) laboratory and details are reported in the Materials and methods under antibody generation and western blotting section. |
| Antibody | Anti-Nuclear Pore Complex Proteins (Mouse monoclonal) | BioLegend | Cat#902902 (MAb414) | IF (1:500) |
| Antibody | Anti-EcR (Mouse monoclonal) | DSHB | Cat#DDA2.7 | IF (1:20) |
| Antibody | Anti-α tubulin (Mouse monoclonal) | DSHB | Cat#12G10 | IB (1:5000) |
| Antibody | Goat anti-rabbit Alexa Fluor Plus 680 | Thermo Fisher Scientific | Cat#A32734 | IB (1:40,000) |
| Antibody | Goat anti-mouse Alexa Fluor Plus 800 | | Cat#A32730 | IB (1:40,000) |
| Antibody | Goat anti-rabbit Alexa Fluor 488 | | Cat#A11034 | IF (1:800) |
| Antibody | Goat anti-rabbit Alexa Fluor 568 | | Cat#A11036 | IF (1:800) |
| Antibody | Goat anti-mouse Alexa Fluor 488 | | Cat#A11029 | IF (1:800) |
| Antibody | Goat anti-mouse Alexa Fluor 568 | | Cat#A11004 | IF (1:800) |
| Antibody | Goat anti-rabbit Alexa Fluor 647 | Jackson ImmunoResearch | Cat#111-605-045 | IF (1:800) |

*Appendix 1 Continued on next page*

*Appendix 1 Continued*

| Reagent type (species) or resource | Designation | Source or reference | Identifiers | Additional information |
|---|---|---|---|---|
| Sequence-based reagent | Nup107_gRNA1 | Generated in the RKM (corresponding authors) laboratory and details are reported in the material and methods under antibody generation and western blotting, Nested PCR, Quantitative RT-PCR, Transgenic fly generation sections | PCR primers (5'–3'): | TGGCCGACAGCCCGTTCCCG |
| Sequence-based reagent | Nup107_ gRNA2 | | PCR primers (5'–3'): | GGAGCTGCTCAACTCGAAACTGG |
| Sequence-based reagent | 5'UTR Primer1_F | | PCR primers (5'–3'): | GCTCCCAAATACTCGCTGCC |
| Sequence-based reagent | 3'UTR Primer1_R | | PCR primers (5'–3'): | CTTCTGCCGGCGGATTTGTT |
| Sequence-based reagent | 5'UTR Primer2_F | | PCR primers (5'–3'): | GGTACCCATACTAATGATTC |
| Sequence-based reagent | 3'UTR Primer2_R | | PCR primers (5'–3'): | CATGTTGTTTGTCTCGCTACT |
| Sequence-based reagent | Nup107 (for antibody generation) | | PCR primers (5'–3'): | Forward: ATCGGATCCATGGCCGACAGCCCGTTC Reverse: ATCGAATTCCTACCACGCCATCATACGATC |
| Sequence-based reagent | Nup107_RT | | PCR primers (5'–3'): | Forward: GCAGGCTCACCGATCGGAAG Reverse: TCCATCTGCAGTAGGCGATG |
| Sequence-based reagent | EcR_RT | | PCR primers (5'–3'): | Forward: AAGAGGATCTCAGGCGTATAA Reverse: GGCCTTTAGTAACGTGATCTG |
| Sequence-based reagent | Eip75A_RT | | PCR primers (5'–3'): | Forward: ACCACAGCACCACCCATTT Reverse: TGTTTGGCGGTAGTTTCAGG |
| Sequence-based reagent | Eip74EF_RT | | PCR primers (5'–3'): | Forward: CTCTGCTCCACATAAAGACG Reverse: CCGCTAAGCAGATTGTGG |
| Sequence-based reagent | Phm_RT | | PCR primers (5'–3'): | Forward: GGATTTCTTTCGGCGCGATGTG Reverse: TGCCTCAGTATCGAAAAGCCGT |
| Sequence-based reagent | Spok_RT | | PCR primers (5'–3'): | Forward: TATCTCTTGGGCACACTCGCTG Reverse: GCCGAGCTAAATTTCTCCGCTT |
| Sequence-based reagent | Dib_RT | | PCR primers (5'–3'): | Forward: TGCCCTCAATCCCTATCTGGTC Reverse: ACAGGGTCTTCACACCCATCTC |
| Sequence-based reagent | Sad_RT | | PCR primers (5'–3'): | Forward: CCGCATTCAGCAGTCAGTGG Reverse: ACCTGCCGTGTACAAGGAGAG |
| Sequence-based reagent | Shd_RT | | PCR primers (5'–3'): | Forward: CGGGCTACTCGCTTAATGCAG Reverse: AGCAGCACCACCTCCATTTC |
| Sequence-based reagent | Torso_RT | | PCR primers (5'–3'): | Forward: CAGCTACTGCGACAAGGTCATCG Reverse: CTCGGTTGCAGCTTGCAGTTG |
| Sequence-based reagent | Rpl49_RT | | PCR primers (5'–3'): | Forward: CGTTTACTGCGGCGAGAT Reverse: GTGTATTCCGACCACGTTACA |
| Commercial assay or kit | RNA isolation kit | Favorgen Biotech | Cat#FATRK-001–2 | |
| Commercial assay or kit | iScriptTM cDNA synthesis kit | Bio-Rad | Cat#170–8891 | |
| Commercial assay or kit | 20-Hydroxyecdysone ELISA kit | Thermo Fisher Scientific | Cat#EIAHYD | |
| Chemical compound, drug | Fluoroshield | Sigma-Aldrich | Cat#F6057 | |
| Chemical compound, drug | Triton X-100 | | Cat#X100-1L | |
| Chemical compound, drug | Tween-20 | | Cat#274348 | |
| Chemical compound, drug | Paraformaldehyde | | Cat#158127 | |
| Chemical compound, drug | iTaq Universal SYBR Green Supermix | Bio-Rad | Cat#1725122 | |
| Chemical compound, drug | G9 Taq Polymerase 10× buffer with MgCl$_2$ | GCC Biotech | Cat#G7115 | |
| Chemical compound, drug | dNTPs | SBS GENETECH | Cat#EN-2 | |
| Chemical compound, drug | 20-Hydroxyecdysone | Sigma-Aldrich | Cat# H5142 | |
| Chemical compound, drug | EDTA | Sigma-Aldrich | Cat#03690 | |
| Chemical compound, drug | SDS | HIMEDIA | Cat#GRM886 | |
| Chemical compound, drug | Proteinase K | MP Biomedicals | Cat#193981 | |
| Chemical compound, drug | Tris | HIMEDIA | Cat#MB029 | |
| Chemical compound, drug | Potassium acetate | HIMEDIA | Cat#MB042 | |
| Chemical compound, drug | Isopropanol | MP Biomedicals | Cat#194006 | |

*Appendix 1 Continued*

| Reagent type (species) or resource | Designation | Source or reference | Identifiers | Additional information |
|---|---|---|---|---|
| Chemical compound, drug | Ethanol | Merck | Cat#100983 | |
| Chemical compound, drug | Sucrose | ANJ Biomedicals | Cat#100314 | |
| Chemical compound, drug | IPTG | Sigma-Aldrich | Cat#I6758 | |
| Chemical compound, drug | Protease inhibitor cocktail (PIC) | Roche | Cat#04693132001 | |
| Chemical compound, drug | NaCl | Emparta ACS | Cat#1.93206.0521 | |
| Chemical compound, drug | EGTA | Sigma-Aldrich | Cat#E4378 | |
| Chemical compound, drug | Sodium deoxycholate | | Cat# 30970 | |
| Chemical compound, drug | Sodium azide | | Cat#438456 | |
| Chemical compound, drug | N-Hydroxy-succinimidyl (NHS) Sepharose | Sigma-Aldrich | Cat#H8280 | |
| Chemical compound, drug | Urea | MP Biomedicals | Cat#194857 | |
| Software, algorithm | ImageJ/Fiji | National Institutes of Health, USA | | http://imagej.nih.gov/ij/ |
| Software, algorithm | GraphPad Prism software | GraphPad | | https://www.graphpad.com/ |
| Software, algorithm | Adobe Photoshop 2023 | Adobe | | https://www.adobe.com/in/products/photoshop.html |
| Software, algorithm | QuantStudio Design & Analysis Software | Applied Biosystems | | https://www.thermofisher.com/in/en/home/products-and-services/promotions.html |

