## [Editor Report · eLife Assessment]

This **valuable** study presents findings on the developmental roles of Nup107, a key nucleoporin, in regulating the larval-to-pupal transition in *Drosophila melanogaster* through its involvement in ecdysone signaling. The evidence supporting the authors' claims is **solid**, with robust experimental approaches including RNAi knockdown and rescue experiments. The authors propose that Nup107 influences EcR localization indirectly by reducing the expression of Halloween genes, a consequence of impaired Torso signaling. However, it remains uncertain whether Torso is the sole receptor tyrosine kinase involved, and this disruption ultimately leads to decreased ecdysone production. In addition, finding a mechanism would strengthen the findings as the currently proposed mechanism is not completely supported by the data.

---

## [Referee Report · Reviewer #1 (Public review)]

This study provides a thorough analysis of Nup107's role in *Drosophila* metamorphosis, demonstrating that its depletion leads to developmental arrest at the third larval instar stage due to disruptions in ecdysone biosynthesis and EcR signaling. Importantly, the authors establish a novel connection between Nup107 and Torso receptor expression, linking it to the hormonal cascade regulating pupariation.

The authors have addressed most of the concerns raised in my initial review, particularly those outlined in the public comments. However, I note that they have not directly responded to several specific points raised in the "Author Recommendations" section. That said, a key mechanistic question remains unresolved and deserves further experimental or at least conceptual clarification.

It has been previously shown that Nup107 regulates the nuclear translocation of dpERK (Kim et al., 2010). This observation may provide a mechanistic explanation for the developmental arrest observed upon Nup107 depletion in the prothoracic gland (PG). Given that PG growth and ecdysone biosynthesis are driven by several receptor tyrosine kinases, it is plausible that loss of Nup107 impairs dpERK nuclear translocation, thereby functionally shutting down RTK-dependent transcriptional responses, including those activating Halloween gene expression. This model is supported by the finding that activated Ras (rasV12) can rescue the arrest, likely by generating sufficiently high levels of dpERK such that some fraction enters the nucleus despite impaired translocation. This hypothesis may explain the discrepancy between the complete developmental arrest observed upon Nup107 depletion and the developmental delay seen in Torso mutants.

Similarly, the rescue by Torso, but not EGFR, may reflect differences in receptor activation thresholds. It has been proposed that Torso overexpression might leads to ligand-independent dimerization and constitutive activity, whereas EGFR overexpression may remain ligand-dependent and thus insufficient under compromised dpERK transport conditions. A critical experiment to validate this model would be to examine dpERK localization in PG cells upon Nup107 depletion. This would help establish whether defective nuclear import of dpERK underlies the observed developmental arrest. Even if technically challenging, the authors should at least discuss this hypothesis explicitly in the revised manuscript.

In addition, it has been shown that TGFβ/Activin signaling regulates Torso expression in the prothoracic gland (PG). Therefore, it is plausible that this pathway may also be affected by impaired nuclear translocation of downstream effectors due to Nup107 depletion. This raises the possibility that Nup107 plays a broad regulatory role, impacting multiple signaling cascades-such as RTK and TGFβ/Activin pathways-by controlling the nuclear import of their key effectors.

---

## [Referee Report · Reviewer #2 (Public review)]

Summary:

The manuscript by Kawadkar et al investigates the role of Nup107 in developmental progression via regulation of ecdysone signaling. The authors identify an interesting phenotype of Nup107 whole body RNAi depletion in *Drosophila* development - developmental arrest at the late larval stage. Nup107-depleted larvae exhibit mis-localization of the Ecdysone receptor (EcR) from the nucleus to the cytoplasm and reduced expression of EcR taret genes in salivary glands, indicative of compromised ecdysone signaling. This mis-localization of EcR in salivary glands was phenocopied when Nup107 was depleted only in the prothoracic gland (PG), suggesting that it is not nuclear transport of EcR but presence of ecdysone (normally secreted from PG) that is affected. Consistently, whole body levels of ecdysone were shown to be reduced in Nup107 KD, particularly at the late third instar stage when a spike in ecdysone normally occurs. Importantly, the authors could rescue the developmental arrest and EcR mis-localization phenotypes of Nup107 KD by adding exogenous ecdysone, supporting the notion that Nup107 depletion disrupts biosynthesis of ecdysone, which arrests normal development. Additionally, they found that rescue of Nup107 KD phenotype can also be achieved by over-expression of the receptor tyrosine kinase torso, which is thought to be the upstream regulator of ecdysone synthesis in the PG. Transcript levels of torso are also shown to be downregulated in the Nup107KD, as are transcript levels of multiple ecdysone biosynthesis genes. Together, these experiments reveal a new role of Nup107 or nuclear pore levels in hormone-driven developmental progression, likely via regulation of levels of torso and torso-stimulated ecdysone biosynthesis.

Strengths:

The developmental phenotypes of an NPC component presented in the manuscript are striking and novel, and the data appears to be of high quality. The rescue experiments are particularly significant, providing strong evidence that Nup107 functions upstream of torso and ecdysone levels in regulation of developmental timing and progression.

Weaknesses:

The underlying mechanism is however not clear, and any insight into how Nup107 may regulate these pathways would greatly strengthen the manuscript. Some suggestions to address this are detailed below.

Major questions:

(1) Determining how specific this phenotype is to Nup107 vs. to reduced NPC levels overall would give some mechanistic insight. Does knocking down other components of the Nup107 subcomplex (the Y-complex) lead to similar phenotypes? Given the published gene regulatory function of Nup107, do other gene regulatory Nups such as Nup98 or Nup153 produce these phenotypes?

(2) In a related issue, does this level of Nup107 KD produce lower NPC levels? It is expected to, but actual quantification of nuclear pores in Nup107-depleted tissues should be added. These and above experiments would help address a key mechanistic question - is this phenotype the result of lower numbers of nuclear pores or specifically of Nup107?

(3) Additional experiments on how Nup107 regulates torso would provide further insight. Does Nup107 regulate transcription of torso or perhaps its mRNA export? Looking at nascent levels of the torso transcript and the localization of its mRNA can help answer this question. Or alternatively, does Nup107 physically bind torso?

(4) The depletion level of Nup107 RNAi specifically in the salivary gland vs. the prothoracic gland should be compared by RT-qPCR or western blotting.

(5) The UAS-torso rescue experiment should also include the control of an additional UAS construct - so Nup107; UAS-control vs Nup107; UAS-torso should be compared in the context of rescue to make sure the Gal4 driver is functioning at similar levels in the rescue experiment.

Minor:

(6) Figures and figure legends can stand to be more explicit and detailed, respectively.

Comments on revisions:

The revised manuscript addresses several outstanding issues, most importantly the question of whether the developmental delay phenotype of Nup107 is exhibited by other Nups.

I recommend that the authors include the data they provide in the rebuttal letter on Nup153 KD not showing the delay phenotype (Figure R1) into the actual manuscript. It's an important mechanistic question raised by multiple reviewers, and would strengthen the authors' conclusions. Ideally, knock downs of other Nups of the Nup107 complex should be investigated, especially given that all those RNAi lines are publicly available.

Figure 6B should also specify whether the torso transcript being measured is mRNA or nascent, as it would help understand whether it's transcription or mRNA stability that is affected by Nup107 KD.

---

## [Referee Report · Reviewer #3 (Public review)]

These findings suggest that Nup107 is involved in regulating ecdysone signaling during developmental transitions, with depletion of Nup107 disrupting hormone-regulated processes. Moreover, the rescue experiments hint that Nup107 might directly influence EcR signaling and ecdysone biosynthesis, though the precise molecular mechanism remains unclear.

Overall, the manuscript presents compelling data supporting Nup107's role in regulating developmental transitions.

Comments on revisions:

RNAi specificity: The authors now provide a more thorough discussion of off-target effects and justify their reliance on the Nup107KK RNAi line. The explanation regarding the predicted off-target for the GD line and their choice to use the KK line with a known insertion site is appropriate and adequately addresses the original concern.

NPC component specificity: The authors clarify that among the Nup107 complex members tested, only Nup107 knockdown induced developmental arrest. Their inclusion of Nup153 as a control helps to support the specificity of the phenotype, although expanding this analysis beyond a single additional Nup would further strengthen the claim.

Mechanistic clarity: The authors now distinguish between Nup107's upstream role in regulating torso and ecdysone biosynthetic genes versus direct control of EcR translocation. The clarification that EcR nuclear localization is 20E-dependent but Nup107-independent improves interpretive clarity.

The molecular mechanism linking Nup107 to torso regulation remains somewhat speculative. A deeper exploration of whether Nup107 influences transcriptional regulation through chromatin association (as the authors suggest) would strengthen the mechanistic narrative.

Conclusion:

Overall, the authors have addressed the major concerns raised in the initial review, and the revised manuscript presents a more coherent and compelling case for Nup107 as a regulator of developmental timing via the ecdysone signaling axis. While some mechanistic questions remain, the core findings are supported by the data, and the work provides novel insights into NPC function in development.

---

## [Author Response]

The following is the authors’ response to the original reviews

**Public Reviews:**

**Reviewer #1 (Public review):**
This study provides a thorough analysis of Nup107's role in *Drosophila* metamorphosis, demonstrating that its depletion leads to developmental arrest at the third larval instar stage due to disruptions in ecdysone biosynthesis and EcR signaling. Importantly, the authors establish a novel connection between Nup107 and Torso receptor expression, linking it to the hormonal cascade regulating pupariation.However, some contradictory results weaken the conclusions of the study. The authors claim that Nup107 is involved in the translocation of EcR from the cytoplasm to the nucleus. However, the evidence provided in the paper suggests it more likely regulates EcR expression positively, as EcR is undetectable in Nup107-depleted animals, even below background levels.

We appreciate the concern raised in this public review. However, we must clarify that we do not claim that Nup107 directly regulates the translocation of EcR from the cytoplasm to nucleus, rather Nup107 regulates Ecdysone hormone (20E) synthesis which in turn affects EcR translocation. In the manuscript, we posited this hypothesis if Nup107 will regulate EcR nuclear translocation (9th line of 2nd paragraph on page 6). We have spelled this out more clearly as the 3rd subsection title of the Results section, and in the discussion (8th line of 2nd paragraph on page 11).

20E acts through the EcR to induce the transcription of EcR responsive genes including the EcR. This creates a positive autoregulatory loop that enhances the EcR level through ecdysone signaling (1). Since Nup107 depletion leads to a reduction in ecdysone levels, it disrupts the transcription autoregulatory EcR expression loop. This can contribute to the reduced EcR levels seen in Nup107-depleted animals.

Additionally, the link between Nup107 and Torso is not fully substantiated. While overexpression of Torso appears to rescue the lack of 20E production in the prothoracic gland, the distinct phenotypes of Torso and Nup107 depletion-developmental delay in the former versus complete larval arrest in the latter complicate understanding of Nup107's precise role.

We understand that there are differences in the developmental delay when Tosro and Nup107 depletion is analyzed. However, the two molecules being compared here are very different, and variability in their depletion could contribute observed phenotypic differences (2). Even if there is no variability of depletion of Torso and Nup107­­­, we believe that Nup107, being more widely expressed, and involved in the regulation of various cellular processes, induces stronger defects.

Further, we think that RNAi-mediated depletion of Nup107 in prothoracic glands (PG) causes significant reduction in the PG size, which may exert a pronounced defect in 20E biosynthesis through the *Halloween* genes, inducing a stronger developmental arrest.

To clarify these discrepancies, further investigation into whether Nup107 interacts with other critical signaling pathways related to the regulation of ecdysone biosynthesis, such as EGFR or TGF-β, would be beneficial and could strengthen the findings.In summary, although the study presents some intriguing observations, several conclusions are not well-supported by the experimental data.

We agree with the reviewer’s suggestion. As noted in the literature, five RTKs-torso, InR, EGFR, Alk, and Pvr-stimulate the PI3K/Akt pathway, which plays a crucial role in the PG functioning and controlling pupariation and body size (3). We have checked the torso and EGFR signaling. We rescued Nup107 defects with the torso overexpression, however, constitutively active EGFR (BL-59843) did not rescue the phenotype (data was not shown). Nonetheless, we plan to examine the EGFR pathway activation by measuring the pERK levels in Nup107-depleted PGs.

**Reviewer #2 (Public review):**
Summary:The manuscript by Kawadkar et al investigates the role of Nup107 in developmental progression via the regulation of ecdysone signaling. The authors identify an interesting phenotype of Nup107 whole-body RNAi depletion in *Drosophila* development - developmental arrest at the late larval stage. Nup107-depleted larvae exhibit mis-localization of the Ecdysone receptor (EcR) from the nucleus to the cytoplasm and reduced expression of EcR target genes in salivary glands, indicative of compromised ecdysone signaling. This mis-localization of EcR in salivary glands was phenocopied when Nup107 was depleted only in the prothoracic gland (PG), suggesting that it is not nuclear transport of EcR but the presence of ecdysone (normally secreted from PG) that is affected. Consistently, whole-body levels of ecdysone were shown to be reduced in Nup107 KD, particularly at the late third instar stage when a spike in ecdysone normally occurs. Importantly, the authors could rescue the developmental arrest and EcR mislocalization phenotypes of Nup107 KD by adding exogenous ecdysone, supporting the notion that Nup107 depletion disrupts biosynthesis of ecdysone, which arrests normal development. Additionally, they found that rescue of the Nup107 KD phenotype can also be achieved by over-expression of the receptor tyrosine kinase torso, which is thought to be the upstream regulator of ecdysone synthesis in the PG. Transcript levels of the torso are also shown to be downregulated in the Nup107KD, as are transcript levels of multiple ecdysone biosynthesis genes. Together, these experiments reveal a new role of Nup107 or nuclear pore levels in hormone-driven developmental progression, likely via regulation of levels of torso and torso-stimulated ecdysone biosynthesis.Strengths:The developmental phenotypes of an NPC component presented in the manuscript are striking and novel, and the data appears to be of high quality. The rescue experiments are particularly significant, providing strong evidence that Nup107 functions upstream of torso and ecdysone levels in the regulation of developmental timing and progression.Weaknesses:The underlying mechanism is however not clear, and any insight into how Nup107 may regulate these pathways would greatly strengthen the manuscript. Some suggestions to address this are detailed below.Major questions:(1) Determining how specific this phenotype is to Nup107 vs. to reduced NPC levels overall would give some mechanistic insight. Does knocking down other components of the Nup107 subcomplex (the Y-complex) lead to similar phenotypes? Given the published gene regulatory function of Nup107, do other gene regulatory Nups such as Nup98 or Nup153 produce these phenotypes?

We thank this public review for raising this concern. Working with a Nup-complex like the Nup107 complex, this concern is anticipated but difficult to address as many Nups function beyond their complex identity. Our observations with all other members of the Nup107-complex, including dELYS, suggest that except Nup107, none of the other tested Nup107-complex members could induce larval developmental arrest.

In this study, we primarily focused on the Nup107 complex (outer ring complex) of the NPC. However, previous studies have reported that Nup98 and Nup153 interact with chromatin, with these investigations conducted in *Drosophila* S2 cells (4, 5, 6). We have now examined other nucleoporins outside of this complex, such as Nup153.

We ubiquitously depleted Nup153 using the *Actin5C-Gal4* driver and assessed the pupariation profile of the knockdown larvae in comparison to control larvae. In contrast to the Nup107 knockdown, when Nup153 is depleted to less than 50% levels, no impact on pupariation was observed (Auhtor response image 1)

**Author response image 1. sa4fig1:** Nup153 depletion does not affect the *Drosophila* metamorphosis.

(2) In a related issue, does this level of Nup107 KD produce lower NPC levels? It is expected to, but actual quantification of nuclear pores in Nup107-depleted tissues should be added. These and the above experiments would help address a key mechanistic question - is this phenotype the result of lower numbers of nuclear pores or specifically of Nup107?

We agree with the concern raised here, and to address the concern raised here, we stained the control and Nup107 depleted salivary glands with mAb414 antibody (exclusively FG-repeat Nup recognizing antibody). While Nup107 intensities are significantly reduced at the nuclear envelope in Nup107 depleted salivary glands, the mAb414 staining seems unperturbed (Author response image 2).

**Author response image 2. sa4fig2:** Nup107 depletion does not perturb overall NPC composition. Comparison of salivary gland nucleus upon control and *Nup107* knockdown. The Nup107 is shown in green and mAb414, staining for other FG-repeat containing nucleoporins is shown in red. Scale bars, 5µm.

(3) Additional experiments on how Nup107 regulates the torso would provide further insight. Does Nup107 regulate transcription of the torso or perhaps its mRNA export? Looking at nascent levels of the torso transcript and the localization of its mRNA can help answer this question. Or alternatively, does Nup107 physically bind the torso?

While the concern regarding torso transcript level is genuine, we have already reported in the manuscript that Nup107 directly regulates torso expression. When Nup107 is depleted, torso levels go down, which in turn controls ecdysone production and subsequent EcR signaling (Figure 6B of the manuscript).

However, the exact nature of Nup107 regulation on torso expression is still unclear. Since the Nup107 is known to interact with chromatin (7), it may affect torso transcription. The possibility of a stable and physiologically relevant interaction between Nup107 and the torso in a cellular context is unlikely largely due to their distinct subcellular localizations. If we investigate this further, it will require a significant amount of time for having reagents and experimentation, and currently stands beyond the scope of this manuscript.

(4) The depletion level of Nup107 RNAi specifically in the salivary gland vs. the prothoracic gland should be compared by RT-qPCR or western blotting.

Although we know that the Nup107 protein signal is reduced in SG upon knockdown (Figure 3B), we have not compared the Nup107 transcript level in these two tissues (SG and PG) upon RNAi. As suggested here, we evaluated the knockdown efficiency of Nup107 using the salivary gland-specific driver *AB1-Gal4* and the prothoracic gland-specific driver *Phm-Gal4*. Our results indicate a significant reduction in Nup107 transcript levels upon Nup107 RNAi in both SG and PG compared to their respective controls (Author response image 3).

**Author response image 3. sa4fig3:** Nup107 levels are significantly reduced upon *Nup107KK* RNAi. Quantification of Nup107 transcript levels from control and Nup107 depleted larvae [tissue specific depletion using *AB1-Gal4* (A) and *Phm*-*Gal4* (B)]. Data are represented from at least three independent experiments. Statistical significance was derived from the Student’s t-test. Error bars represent SEM. **p = <0.004

(5) The UAS-torso rescue experiment should also include the control of an additional UAS construct - so Nup107; UAS-control vs Nup107; UAS-torso should be compared in the context of rescue to make sure the Gal4 driver is functioning at similar levels in the rescue experiment.

This is a very valid point, and we took this into account while planning the experiment. In such cases, often the GAL4 dilution can be critical. We have demonstrated in Figure S7, that GAL4 dilution is not blurring our observations. We used the Nup107^KK^; UAS-GFP as control alongside the Nup107^KK^; UAS-torso. We conclude that the presence of GFP signals in prothoracic glands and their reduced size indicates genes downstream to both UAS sequences are transcribed, and GAL4 dilution does not play a role here.

Minor:(6) Figures and figure legends can stand to be more explicit and detailed, respectively.

We have revisited all figures and their corresponding legends to ensure appropriate and explicit details are provided.

**Reviewer #3 (Public review):**
Summary:In this study by Kawadkar et al, the authors investigate the developmental role of Nup107, a nucleoporin, in regulating the larval-to-pupal transition in *Drosophila* through RNAi knockdown and CRISPR-Cas9-mediated gene editing. They demonstrate that Nup107, an essential component of the nuclear pore complex (NPC), is crucial for regulating ecdysone signaling during developmental transitions. The authors show that the depletion of Nup107 disrupts these processes, offering valuable insights into its role in development.Specifically, they find that:(1) Nup107 depletion impairs pupariation during the larval-to-pupal transition.(2) RNAi knockdown of Nup107 results in defects in EcR nuclear translocation, a key regulator of ecdysone signaling.(3) Exogenous 20-hydroxyecdysone (20E) rescues pupariation blocks, but rescued pupae fail to close.(4) Nup107 RNAi-induced defects can be rescued by activation of the MAP kinase pathway.Strengths:The manuscript provides strong evidence that Nup107, a component of the nuclear pore complex (NPC), plays a crucial role in regulating the larval-to-pupal transition in *Drosophila*, particularly in ecdysone signaling.The authors employ a combination of RNAi knockdown, CRISPR-Cas9 gene editing, and rescue experiments, offering a comprehensive approach to studying Nup107's developmental function.The study effectively connects Nup107 to ecdysone signaling, a key regulator of developmental transitions, offering novel insights into the molecular mechanisms controlling metamorphosis.The use of exogenous 20-hydroxyecdysone (20E) and activation of the MAP kinase pathway provides a strong mechanistic perspective, suggesting that Nup107 may influence EcR signaling and ecdysone biosynthesis.Weaknesses:The authors do not sufficiently address the potential off-target effects of RNAi, which could impact the validity of their findings. Alternative approaches, such as heterozygous or clonal studies, could help confirm the specificity of the observed phenotypes.

This is a very valid point raised, and we are aware of the consequences of the off-target effects of RNAi. To assert the effects of authentic RNAi and reduce the off-target effects, we have used two RNAi lines (*Nup107GD* and *Nup107KK*) against Nup107. Both RNAi induced comparable levels of Nup107 reduction, and using these lines, ubiquitous and PG specific knockdown produced similar phenotypes. Although the *Nup107GD* line exhibited a relatively stronger knockdown compared to the *Nup107KK* line, we preferentially used the *Nup107KK* line because the *Nup107GD* line is based on the P-element insertion, and the exact landing site is unknown. Furthermore, there is an off-target predicted for the *Nup107GD* line, where a 19bp sequence aligns with the bifocal (*bif*) sequence. The *bif*-encoded protein is involved in axon guidance and regulation of axon extension. However, the *Nup107KK* line does not have a predicted off-target molecule, and we know its precise landing site on the second chromosome. Thus, the *Nup107KK* line was ultimately used in experimentation for its clearer and more reliable genetic background.

We are also investigating Nup107 knockdown in the prothoracic gland, which exhibits polyteny. Additionally, the number of cells in the prothoracic gland is quite limited, approximately 50-60 cells (8). Given this, there is a possibility that a clonal study may not yield the phenotype.

NPC Complex Specificity: While the authors focus on Nup107, it remains unclear whether the observed defects are specific to this nucleoporin or if other NPC components also contribute to similar defects. Demonstrating similar results with other NPC components would strengthen their claims.

We thank this public review for raising this concern. Working with a Nup-complex like the Nup107 complex, this concern is anticipated but difficult to address as many Nups function beyond their complex identity. Our observations with all other members of the Nup107-complex, including *dELYS*, suggest that except *Nup107*, none of the other Nup107-complex members could induce larval developmental arrest. Since the study is primarily focused on the Nup107 complex (outer ring complex) of the NPC, we have not examined many more nucleoporins outside of this complex. But our observations with Nup153 knockdown, a nuclear basket nucleoporin, is comparable to control, with no delay in development (Author response image 1)

Although the authors show that Nup107 depletion disrupts EcR signaling, the precise molecular mechanism by which Nup107 influences this process is not fully explored. Further investigation into how Nup107 regulates EcR nuclear translocation or ecdysone biosynthesis would improve the clarity of the findings.

We appreciate the concern raised. Through our observation, we have proposed the upstream effect of Nup107 on the PTTH-torso-20E-EcR axis regulating developmental transitions. We know that Nup107 regulates torso levels, but we do not know if Nup107 directly interacts with torso. We would like to address whether Nup107 exerts control on PTTH levels also.

However, we must emphasize that Nup107 does not directly regulate the translocation of EcR. On the contrary, we have demonstrated that when Nup107 is depleted only in the salivary gland, EcR translocates into the nucleus. Thus we conclude that the EcR translocation is 20E dependent and Nup107 independent. Further, we have argued that Nup107 regulates the expression of *Halloween* genes required for ecdysone biosynthesis. We are interested in identifying if Nup107 associates directly or through some protein to chromatin to bring about the changes in gene expression required for normal development.

There are some typographical errors and overly strong phrases, such as "unequivocally demonstrate," which could be softened. Additionally, the presentation of redundant data in different tissues could be streamlined to enhance clarity and flow.

Response: We thank the reviewer for this observation. We have put our best efforts to remove all typographical errors and have now made more reasonable statements based on our conclusions.

**Recommendations for the Authors:**

**Reviewer #1 (Recommendations for the authors):**
The manuscript presents compelling evidence that Nup107 plays a role in regulating ecdysone production. However, significant concerns remain regarding the effects on EcR localization and expression, as well as the claimed link between PTTH/Torso signaling and Nup107's function, as the evidence provided is not conclusive.The hypothesis that Nup107 mediates EcR translocation from the cytoplasm to the nucleus appears misinterpreted by the authors. Based on the presented images, particularly for the prothoracic gland (PG) Figure 3C, Nup107 depletion seems to impact EcR protein levels rather than its localization. This conclusion is supported by data showing that EcR transcripts are autonomously downregulated in the absence of Nup107. Furthermore, the restoration of nuclear EcR levels upon exogenous 20E supplementation suggests that (1) Nup107 is dispensable for EcR activation and function, and (2) its primary role lies in regulating ecdysone production.

We appreciate the concern raised by reviewer. However, we must clarify that we do not claim that Nup107 directly regulates the translocation of EcR from the cytoplasm, rather Nup107 regulates Ecdysone hormone (20E) synthesis which in turn affects EcR translocation. In the manuscript, we posited this hypothesis if Nup107 will regulate EcR nuclear translocation (9th line of 2nd paragraph on page 6). We have spelled this out more clearly as the 3rd subsection title of the Results section, and in the discussion (8th line of 2nd paragraph on page 11).

20E acts through the EcR to induce the transcription of EcR responsive genes including the EcR. This creates a positive autoregulatory loop that enhances the EcR level through ecdysone signaling (1). Since Nup107 depletion leads to a reduction in ecdysone levels, it disrupts the transcription autoregulatory EcR expression loop. This can contribute to the reduced EcR levels seen in Nup107-depleted animals.

Given that nucleoporins are known to influence mRNA transport-for instance, Nup107 has been shown to control Scn5a mRNA transport (Guan et al., 2019)-the observed effects on Halloween gene and EcR expression may stem from disruptions in mRNA transport to the cytoplasm. The downregulation of Shade further supports this hypothesis, as restricted ecdysone biosynthesis typically induces Shade upregulation in peripheral tissues. Quantifying potential mRNA accumulation in the nuclei of PG cells in Nup107-depleted animals would clarify this.

The reviewer raised a valid point, and we fully agree with the concern that Nup107 has been shown to control Scn5a mRNA transport (Guan et al., 2019). The observed effects on Halloween gene and EcR expression could indeed stem from disruptions in efficient mRNA export to the cytoplasm. However, if Nup107 were regulating the mRNA export of Halloween genes and EcR, we should not expect a rescue of the Nup107 developmental delay phenotype with torso overexpression. But, by overexpressing the torso in the Nup107 depletion background, we are activating the torso pathway dependent Halloween gene expression, and rescuing the developmental delay phenotype of Nup107 depletion.

With the current data, it is difficult to conclusively claim a role for Nup107 in EcR translocation or expression. Additional experiments, such as EcR overexpression in Nup107-depleted animals or Nup107 overexpression, would help determine its precise role.

We appreciate the concern raised by reviewer. We did attempt to rescue the Nup107 depletion phenotype by overexpressing EcR (BL-6868) in the Nup107-RNAi background. However, we were unable to rescue the Nup107 depletion dependent developmental delay phenotype with this approach. This further suggests that the phenotype is not merely due to low level of EcR, but it is due to low availability of ecdysone hormone and EcR signaling.

The second major issue is the proposed link between Nup107 and PTTH/Torso signaling. The authors suggest that Nup107 regulates ecdysone production through Torso expression based on rescue experiments. However, this is inconsistent with the distinct phenotypes observed when Nup107 or Torso signaling is disrupted. While PTTH/Torso signaling causes only a modest developmental delay (12 hours to 2 days, depending on the mutant), Nup107 depletion results in a complete developmental arrest at the larval stage. This discrepancy raises doubts about the assertion that Torso overexpression alone rescues such a severe phenotype. One possibility is that PTTH levels are upregulated in Nup107-depleted animals, leading to overactivation of the pathway when Torso is overexpressed. Quantifying PTTH levels in Nup107-depleted animals could address this.

The reviewer raised a valid point, and we fully acknowledge this concern. While we do not completely agree with the idea of PTTH upregulation in Nup107 depleted larvae, as suggested here, we believe that quantifying PTTH levels upon Nup107 depletion can provide a useful insight. To address it, we quantified PTTH levels in Nup107-depleted larvae and found no significant change in PTTH expression compared to controls (Author response image 4).

**Author response image 4. sa4fig4:** Nup107 knockdown does not affect the PTTH level. Quantitation of *PTTH* transcript levels from control and Nup107 depleted larvae (Prothoracic specific depletion *Phm-Gal4*). Data are represented from at least three independent experiments. Statistical significance was derived from the Student's t-test. ns is non-significant.

Another possibility is that the stock used for Torso overexpression, which includes a trk mutant, may introduce genetic interactions that overactivate the pathway. Using a clean UAS-Torso stock would resolve this issue.

We appreciate the reviewer’s observation regarding the use of the Torso overexpression line (BL-92604), which carries the *trk* null allele on the second chromosome. The cleaved form of the *trk* serves as ligand for the troso receptor. Since it may serve as ligand for the torso, I am not sure how *trk* null allele bearing line when used along for torso overexpression studies will overactivate the pathway.

We realized this concern and the fly line used in this study and reported in the manuscript was generated through the following genetic strategy using the BL-92604 line. First, a double balancer stock (*Sco/CyO*; *MKRS/TM6.Tb*) was used to generate the *Sco/CyO; UAS-torso/ UAS-torso* genotype. This recombinant line was subsequently combined with the *Nup107KK* line. Through the use of the double balancer strategy, we effectively replaced Nup107 RNAi genotype on the second chromosome, thereby ensuring that our final experimental setup is free from *trk* mutant contamination, if at all.

Moreover, the rescue of Nup107 depletion phenotypes by RasV12 overexpression suggests that multiple RTKs, not just Torso, are affected. EGFR signaling, the primary regulator of ecdysone biosynthesis in the PG during the last larval stage, is notably absent from the authors' analysis. EGFR inactivation is known to arrest development, and previous studies indicate that Nup107 can reduce EGFR pathway activity (Kim et al, 2010). The authors should analyze EGFR pathway activity in the absence of Nup107. Overexpressing EGF ligands like Vein or Spitz in the PG (rather than the receptor) in a Nup107-depleted background would provide more relevant insights.

The RasGTPase is one of the common effector molecules downstream of an activated receptor kinase. Rescue with a constitutively activated form of RasGTPase (RasV12) suggests one of the routes which is activated downstream of the torso receptor. It does not directly suggest all different RTKs are affected and are involved. Our idea of performing a rescue experiment was to see if the pathway activated downstream of the torso involves RasGTPase.

As noted in the literature, five RTKs—torso, InR, EGFR, Alk, and Pvr—stimulate the PI3K/Akt pathway, which plays a crucial role in the PG for controlling pupariation and body size (3). Although EGFR signaling is important, PTTH/Torso signaling is considered the primary mediator of metamorphic timing. In response to the suggestion to analyze EGFR pathway activity in the absence of Nup107, we attempted to rescue the phenotype by overexpressing constitutively active EGFR (BL-59843) in the Nup107-depleted background (data was not shown). We used constitutively active EGFR to bypass the availability of its ligands (vein and spitz). Unfortunately, we were unable to rescue the phenotype with this approach, which further suggests that EGFR is not the targeted RTK pathway in this context. By rescuing with torso, we found that Nup107 regulates torso-mediated Ras/Erk signaling to control metamorphosis.

Additional issues require clarification:(1) RNAi Efficiency: In Figure 1C, the Nup107GD line shows a stronger knockdown effect than Nup107KK, yet most experiments were conducted with the weaker line. This might explain the residual Nup107 protein observed in Figure 2. Could the authors justify this choice?

This is a very valid point raised, and we are aware of the consequences of the off-target effects of RNAi. To assert the effects of authentic RNAi and reduce the off-target effects, we have used two RNAi lines (*Nup107GD* and *Nup107KK*) against Nup107. Both RNAi induced comparable levels of Nup107 reduction, and using these lines, ubiquitous and PG specific knockdown produced similar phenotypes. Although the *Nup107GD* line exhibited a relatively stronger knockdown compared to the *Nup107KK* line, we preferentially used the *Nup107KK* line because the *Nup107GD* line is based on the P-element insertion, and the exact landing site is unknown. Furthermore, there is an off-target predicted for the *Nup107GD* line, where a 19bp sequence aligns with the bifocal (*bif*) sequence. The *bif*-encoded protein is involved in axon guidance and regulation of axon extension. However, the *Nup107KK* line does not have a predicted off-target molecule, and we know its precise landing site on the second chromosome. Thus, the *Nup107KK* line was ultimately used in experimentation for its clearer and more reliable genetic background.

(2) Control Comparisons: In Figure 3, the effects of Nup107 depletion on EcR expression in salivary glands (SG) and PG are shown, but only SG controls are provided. Including PG controls would enable proper comparisons. These controls should also be added to Figures 5, 6, and S5.

As suggested by the reviewer, we have checked the EcR localization in prothoracic gland (Author response image 5), also. As shown in figure R5, when PGs isolated from control, Nup107-RNAi and torso overexpression in Nup107 background were stained for EcR, the observations made were indistinguishable from those made in SGs of the indicated genetic combinations. This indicated that Nup107 regulates EcR signaling by regulating the 20E biosynthesis.

**Author response image 5. sa4fig5:** Prothoracic gland’s specific torso expression rescues EcR nuclear translocation defects. Immunofluorescence-based detection of nucleocytoplasmic distribution of EcR (EcR antibody, red) in control, prothoracic gland specific Nup107 knockdown (*Phm-Gal4>Nup107KK*) and torso overexpressing PG-specific Nup107 knockdown (*Phm-Gal4>Nup107KK*; *UAS-torso*) third instar larval Prothoracic gland nuclei. DNA is stained with DAPI. Scale bars, 20 μm.

(3) Clarify the function of Torso in the text: The authors must revise their description of Torso signaling as the primary regulator of ecdysone production in both the results and discussion sections. Specifically, in the results section, the claim that Torso depletion induces developmental arrest is inaccurate. Instead, available evidence, including Rewitz et al. 2009, demonstrates that Torso depletion causes a delay of approximately five days rather than a complete developmental arrest. This discrepancy should be corrected to avoid overstating the role of Torso signaling in ecdysone regulation and to align the manuscript with established findings.

We agree with the reviewer. We have incorporated the suggestion at the relevant place in the main manuscript.

**Reviewer #3 (Recommendations for the authors):**
These findings suggest that Nup107 is involved in regulating ecdysone signaling during developmental transitions, with depletion of Nup107 disrupting hormone-regulated processes. Moreover, the rescue experiments hint that Nup107 might directly influence EcR signaling and ecdysone biosynthesis, though the precise molecular mechanism remains unclear.Overall, the manuscript presents compelling data supporting Nup107's role in regulating developmental transitions. However, I have a few comments for consideration:Major Comments:RNAi Specificity: While RNAi is a powerful tool, the authors do not sufficiently address potential off-target effects, which could undermine the conclusions. Although a mutant Nup107 is described, it is lethal-are heterozygous or clonal studies possible to validate the findings more robustly?

This is a very valid point raised, and we are aware of the consequences of the off-target effects of RNAi. To assert the effects of authentic RNAi and reduce the off-target effects, we have used two RNAi lines (*Nup107GD* and *Nup107KK*) against Nup107. Both RNAi induced comparable levels of Nup107 reduction, and using these lines, ubiquitous and PG specific knockdown produced similar phenotypes. Although the *Nup107GD* line exhibited a relatively stronger knockdown compared to the *Nup107KK* line, we preferentially used the *Nup107KK* line because the *Nup107GD* line is based on the P-element insertion, and the exact landing site is unknown. Furthermore, there is an off-target predicted for the *Nup107GD* line, where a 19bp sequence aligns with the bifocal (*bif*) sequence. The *bif*-encoded protein is involved in axon guidance and regulation of axon extension. However, the *Nup107KK* line does not have a predicted off-target molecule, and we know its precise landing site on the second chromosome. Thus, the *Nup107KK* line was ultimately used in experimentation for its clearer and more reliable genetic background.

Following the suggestion from the reviewer, we considered conducting heterozygous and clonal analyses using the Nup107 mutant. We have carried out Nup107 knockdown studies in the prothoracic gland, which has a limited number of cells (50-60 cells) and is known to exhibit polyteny (8). Keeping these aspects of the Prothoracic gland in mind, the possibility that a clonal study will yield the phenotype is scarce. However, we will consider moving forward with this approach also.

(2) NPC Complex Specificity: It remains unclear whether the observed defects are specific to Nup107 or if other NPC components also cause similar defects. If the authors are unable to use Nup107 mutants, they could demonstrate similar defects with other critical NPC members to bolster their claim.

We thank this public review for raising this concern. Working with a Nup-complex like the Nup107 complex, this concern is anticipated but difficult to address as many Nups function beyond their complex identity. Our analysis of Nup153 depleted organisms indicates no developmental delay/defect. We have also assessed effects of knockdown of all other members of the Nup107-complex, including dELYS, but except Nup107 no other member of the Nup107-complex could induce developmental arrest in the third instar stage causing lack of pupariation. However, the null mutant of Nup133, the direct interactor of Nup107 in the Nup107-complex, induces a delay in pupariation (unpublished data).

(3) Molecular Mechanism of EcR Signaling: The manuscript shows that Nup107 depletion affects EcR signaling and ecdysone biosynthesis, but the molecular basis of this regulation is not fully explored. Does phosphorylated ERK (p-ERK) fail to enter the nucleus? Clarifying this mechanism would strengthen the study's impact.

We appreciate the reviewer’s insightful comment and fully agree with the concern. To address this, we examined the subcellular localization of phosphorylated ERK (p-ERK) in the prothoracic gland of control larvae, Nup107-depleted larvae, and Nup107-depleted larvae with torso overexpression. In control larvae, p-ERK was predominantly localized in the nucleus. However, in Nup107-depleted larvae, p-ERK was largely retained in the cytoplasm, indicating impaired pathway activation and nuclear translocation. Notably, overexpression of the torso in the Nup107-depleted background restored nuclear localization of p-ERK in the prothoracic gland (Author response image 6). These findings suggest that Nup107 regulates *Drosophila* metamorphosis, in part, through modulation of torso-mediated MAPK signaling.

**Author response image 6. sa4fig6:** Nup107 regulates torso activation dependent p-ERK localization. Detection of nucleocytoplasmic distribution of p-ERK (anti- p-ERK antibody, green) in the third instar larval prothoracic glands of control, PG-specific Nup107 knockdown (*Phm-Gal4>Nup107KK*) and PG-specific torso overexpression in Nup107 knockdown background (*Phm-Gal4>Nup107KK*; *UAS-torso*). DNA is stained with DAPI. Scale bars, 20 µm.

Minor Comments:(1) The manuscript contains typographical errors that may hinder readability. Additionally, some phrases (e.g., "unequivocally demonstrate") may be overly strong. Consider adjusting language to reflect the nature of the data more accurately.

We agree with the reviewer. We have edited the manuscript accordingly to crease out such typographical errors at relevant places in the main manuscript.

(2) The data presentation could be improved by eliminating redundancy. Some sections repeat similar findings in different tissues, which could be consolidated to improve clarity and flow.

While we agree with the comment, we could not help ourselves in tissue redundancy for presenting our data for EcR translocation studies. I wish we could use another tissue. However, we have put EcR localization and p-ERK translocation data in the responses to present another non-redundant tissue perspective (Figures R5 and R6).

References:

(1) Varghese, Jishy, and Stephen M Cohen. “microRNA miR-14 acts to modulate a positive autoregulatory loop controlling steroid hormone signaling in *Drosophila*.” Genes & development vol. 21,18 (2007): 2277-82. doi:10.1101/gad.439807

(2) Rewitz, Kim F et al. “The insect neuropeptide PTTH activates receptor tyrosine kinase torso to initiate metamorphosis.” Science (New York, N.Y.) vol. 326,5958 (2009): 1403-5. doi:10.1126/science.1176450

(3) Pan, Xueyang, and Michael B O'Connor. “Coordination among multiple receptor tyrosine kinase signals controls *Drosophila* developmental timing and body size.” Cell reports vol. 36,9 (2021): 109644. doi:10.1016/j.celrep.2021.109644

(4) Pascual-Garcia, Pau et al. “Metazoan Nuclear Pores Provide a Scaffold for Poised Genes and Mediate Induced Enhancer-Promoter Contacts.” Molecular cell vol. 66,1 (2017): 63-76.e6. doi:10.1016/j.molcel.2017.02.020

(5) Pascual-Garcia, Pau et al. “Nup98-dependent transcriptional memory is established independently of transcription.” eLife vol. 11 e63404. 15 Mar. 2022, doi:10.7554/eLife.63404

(6) Kadota, Shinichi et al. “Nucleoporin 153 links nuclear pore complex to chromatin architecture by mediating CTCF and cohesin binding.” Nature communications vol. 11,1 2606. 25 May. 2020, doi:10.1038/s41467-020-16394-3

(7) Gozalo, Alejandro et al. “Core Components of the Nuclear Pore Bind Distinct States of Chromatin and Contribute to Polycomb Repression.” Molecular cell vol. 77,1 (2020): 67-81.e7. doi:10.1016/j.molcel.2019.10.017

(8) Shimell, MaryJane, and Michael B O'Connor. “Endoreplication in the *Drosophila melanogaster* prothoracic gland is dispensable for the critical weight checkpoint.” microPublication biology vol. 2023 10.17912/micropub.biology.000741. 21 Feb. 2023, doi:10.17912/micropub.biology.000741